# Inflammation in Prostate Cancer: Exploring the Promising Role of Phenolic Compounds as an Innovative Therapeutic Approach

**DOI:** 10.3390/biomedicines11123140

**Published:** 2023-11-24

**Authors:** Raquel Fernandes, Cátia Costa, Rúben Fernandes, Ana Novo Barros

**Affiliations:** 1Centre for Research and Technology of Agro-Environmental and Biological Sciences, CITAB, Inov4Agro, University of Trás-os-Montes and Alto Douro, UTAD, Quinta de Prados, 5000-801 Vila Real, Portugal; catiac@utad.pt; 2FP-I3ID, Instituto de Investigação, Inovação e Desenvolvimento, FP-BHS, Biomedical and Health Sciences, Universidade Fernando Pessoa, 4249-004 Porto, Portugal; ruben.fernandes@ufp.edu.pt; 3CECLIN, Centro de Estudos Clínicos, Hospital Fernando Pessoa, 4420-096 Gondomar, Portugal; 4I3S, Instituto de Investigação e Inovação em Saúde, Universidade do Porto, 4200-135 Porto, Portugal

**Keywords:** prostate cancer, inflammation, immune response, leukocytes, polyphenol compounds, polyphenol-based nanoparticles

## Abstract

Prostate cancer (PCa) remains a significant global health concern, being a major cause of cancer morbidity and mortality worldwide. Furthermore, profound understanding of the disease is needed. Prostate inflammation caused by external or genetic factors is a central player in prostate carcinogenesis. However, the mechanisms underlying inflammation-driven PCa remain poorly understood. This review dissects the diagnosis methods for PCa and the pathophysiological mechanisms underlying the disease, clarifying the dynamic interplay between inflammation and leukocytes in promoting tumour development and spread. It provides updates on recent advances in elucidating and treating prostate carcinogenesis, and opens new insights for the use of bioactive compounds in PCa. Polyphenols, with their noteworthy antioxidant and anti-inflammatory properties, along with their synergistic potential when combined with conventional treatments, offer promising prospects for innovative therapeutic strategies. Evidence from the use of polyphenols and polyphenol-based nanoparticles in PCa revealed their positive effects in controlling tumour growth, proliferation, and metastasis. By consolidating the diverse features of PCa research, this review aims to contribute to increased understanding of the disease and stimulate further research into the role of polyphenols and polyphenol-based nanoparticles in its management.

## 1. Introduction

The prostate, a walnut-shaped gland in males, plays a vital role in the production of seminal fluid, which serves to nourish and transport sperm [1]. Several risk factors including age, ethnicity, genetic predisposing, infection, obesity, and diet, have been linked to the development of prostate malignancies, subsequently leading to prostate inflammation and carcinogenesis [2]. Prostate cancer (PCa) stands as a significant cause of morbidity and cancer-related deaths among men [3]. As the population ages and the prevalence of food processing, coupled with poor dietary habits, continues to rise, there is an anticipated increase in the absolute number of PCa cases [4,5]. However, it is crucial to note that the association between a high-fat diet and obesity as risk factors for PCa development remains a subject of controversy, a topic that will be explored in the following sections. Inflammation is a pivotal player in the development of prostate carcinogenesis. Dysregulation in the mechanisms governing the production and activation of inflammatory cells contributes to abnormal damage within the prostate tissue. Furthermore, the prostate tumour microenvironment hosts highly heterogeneous and plastic cell populations, which leads to patient resistance to therapies and heightened disease recurrence [6]. PCa cells may be modulated into different phenotypes in response to different signals. Consequently, immune cells such as neutrophils, basophils, eosinophils, mast cells, macrophages, and B and T lymphocytes within the tumour microenvironment can be activated into either pro-tumoral or anti-tumoral phenotypes. The precise roles of each immune population in PCa progression, as well as the controversial aspects surrounding the inflammatory mediators they produce, remain subjects of debate in the literature. In most reviewing articles available, the primary focus is on immune populations with higher prevalence in PCa and their roles in promoting cancer progression through pro-inflammatory mediators. However, it is crucial to recognize the importance of other immune populations that influence the phenotype of tumour-associated infiltrates. Additionally, the specific mechanisms governing cytokine and chemokine production from distinct immune populations, along with their roles in particular pathways, are areas where gaps in knowledge persist.

Recent investigations have explored the role of polyphenol compounds and their incorporation into nanoparticles in the context of PCa [7,8,9]. Data suggests that their capacity to scavenge free radicals and act as antioxidants could hold promise for improving PCa therapies. This article seeks to delve into the roles of various leukocyte populations, their production of inflammatory mediators, and their relationships with PCa progression. It will also provide a comprehensive exploration of the role of polyphenols in PCa development and progression, shedding light on potential avenues for their incorporation into nanoparticles. This innovative approach aims to enhance polyphenol delivery to target cells, thereby increasing the effectiveness of current therapies while reducing side effects and therapy resistance.

## 2. Epidemiology of Prostate Cancer

According to the World Health Organization (WHO) [10] the global incidence of PCa was 1,414,259 cases in 2020, with 375,304 reported deaths. PCa incidence varies significantly across different geographic regions and among ethnic groups. Incidence rates range from 6.3 to 83.4 cases per 100,000 people worldwide [11]. The countries with the highest PCa incidence are Northern Europe, Western Europe, and the Caribbean, while the highest mortality rates are observed in the Caribbean, Middle Africa, and Southern Africa [10]. Notably, Black men are more susceptible to PCa than White men, with a higher risk of aggressive carcinogenesis and mortality. These disparities may be attributed to factors such as mistrust of the healthcare system, lack of education, information, and access to diagnosis and treatment, as well as societal stigma associated with the disease [12]. Conversely, the high PCa incidence rates observed in developed countries can be attributed to proactive diagnosis and prevention measures established within healthcare systems. This includes the widespread practice of prostate-specific antigen (PSA) testing for screening [13]. Looking ahead to the next decade, the aging global population is expected to drive an increase in PCa cases to an estimated 1.7 million new cases and 499,000 deaths [14].

## 3. Diagnosis of Prostate Cancer

PCa is typically asymptomatic, which means that by the time it is clinically detected, it has usually reached an advanced stage and often metastasized to other organs. Consequently, clinical therapies tend to be ineffective at this stage, resulting in high mortality rates associated with PCa. Given this scenario, there was a pressing need to implement screening measures for this disease to diagnose it at a treatable stage. This need led to the discovery of the of PSA [15]. After its clinical implementation, the PSA test allowed for the detection of more cases of PCa, leading to an increase in its incidence [16,17]. In fact, the European Randomized Study of Screening for Prostate Cancer (ERSPC) reported a 20% reduction of PCa mortality following PSA screening. However, this screening also led to overdiagnosis, causing increased anxiety due to false positive PSA tests, as well as complications related to further biopsies and hospitalizations [18].

According to the National Comprehensive Cancer Network (NNCN) Foundation, the clinical approach and guidelines for patient diagnosis depend on the stage of the disease [19]. If PSA levels are higher than normal for a patient’s age, it is recommended to perform additional imaging tests, biopsies, or genetic tests.

PSA, a serine protease produced in the prostate epithelium and overexpressed in PCa tissues, is widely used as a screening test for PCa diagnosis [20]. Normal PSA levels also vary with age. NNCCN defines normal PSA ranges as 0.0–2.5 ng/mL, 2.5–3.5 ng/mL, 3.5–4.5 ng/mL, and 4.5–6.5 ng/mL for patients between 40 and 49 years, 50 and 59 years, 60 and 69 years, and 70 and 79 years, respectively. In clinical practice, the PSA test is the most common screening method for PCa. If PSA levels are elevated but the patient exhibits no other symptoms of PCa, a second PSA test is recommended. Another used screening method is the digital rectal exam (DRE), which is a straightforward way to assess the size and texture of the prostate. Typically, this test is used in conjunction with the PSA test, taking into consideration factors such as age, race, or family history of PCa. If PSA levels are higher than normal or if patients have risk factors such as a family history, race, or age, suggesting a potential case of PCa, additional diagnostic tests become necessary (Figure 1).

PCa stages and cell patterns are defined by Gleason Patterns, which are used to estimate the tumour’s Gleason Score. According to the Gleason Score, the tumour’s classification is translated into a tumour grade group that estimates the risk of PCa. The Gleason Pattern ranges from 3 to 5, where 3 resembles normal cells, and 5 is attributed to cells with an abnormal pattern [21]. PCa cells exhibit significant heterogeneity, so the Gleason Pattern considers a primary pattern related to the pattern of cells found in the largest area of the tumour and a secondary pattern accounting for the second largest area [19]. The Gleason Score is the sum of these primary and secondary Gleason Patterns. Gleason Scores ranging from 2 (1 + 1) to 5 (3 + 2) are considered benign, with only tumours scoring 6 (3 + 3) or higher classified as malignant. A higher Gleason Score indicates that the tumour is more likely to grow and spread rapidly. Prostate tumours assigned a Gleason Score of 6 (3 + 3) or 7 (3 + 4 or 4 + 3) are considered low-grade and intermediate-grade, respectively, while tumours with a score of 8 (4 + 4, 3 + 5, or 5 + 3), 9 (4 + 5 or 5 + 4), or 10 (5 + 5) are categorized as high-grade [22]. Following this, a Grade Group, numbered 1 to 5, is assigned based on the Gleason Score, reflecting tumour aggressiveness. Grade Group 1 corresponds to a Gleason Score of 6 and represents the lowest score. Grade Group 2 (3 + 4) and 3 (4 + 4) correspond to a Gleason Score of 7 while Grade Groups 4 and 5 correspond to Gleason Scores of 8 and 9 or 10, respectively, representing the highest level of malignancy. Histologically, Grade Group 1 is characterized by individual, discrete, and well-formed glands. Grade Group 2 has predominantly well-formed glands with a few poorly-formed/fused/cribriform glands. Grade Group 3 has predominantly poorly formed/fused/cribriform glands with lesser (5%) component of well-formed glands. Grade Group 4 has only poorly formed/fused/cribriform glands (4 + 4), or predominantly well-formed glands and lesser component lacking glands (3 + 5), or predominantly lacking glands and lesser component of well-formed glands (5 + 3). Grade Group 5 has lack gland formation (or with necrosis) with or without poorly-formed/fused/cribriform glands (Table 1) [23].

## 4. Pathophysiology of Prostate Cancer

The prostate comprises a central zone (CZ) that contains the ductal tube from the seminal vesicle, the peripheral zone (PZ) located at the posterior region and where the majority of cancer appear, and the transitional zone (TZ) placed below the bladder [1]. The prostate consists of organized layers with three types of epithelial cells—basal, luminal, and neuroendocrine cells—a fibro-muscular network, an endothelial membrane, and immune cells [27]. Basal cells constitute 40% of the epithelium and are characterized by the expression of cytokeratin (KRT) 5, KRT14, KRT17, and p63. Luminal cells make up 60% of the total epithelium and express KRT8, KRT18, cluster of differentiation (CD) 26 and androgen-regulated secretory proteins such as kallikrein related peptidase 3 (KLK3). Neuroendocrine cells are found in lower percentages in the basal lamina, approximately 1%, and express chromogranin A (CHGA) [27,28] (Figure 2a,b).

Disruptions in the epithelial lineages, alterations in the number and phenotype of epithelial cells [29], along with mutations in tumour suppressors, oncogenes [30] and external factors that induce inflammation [2] can result in the dysregulation of the prostate environment. These events may lead to abnormal production of epithelial cells, including an overproduction of luminal cells and a decreased production of basal cells, constituting 99% and 0.1% of tumours, respectively. Simultaneously, there is a breakdown of the basement membrane, infiltration of immune cells, and increased stromal reactivity. Smooth muscle cells are replaced by activated fibroblasts and myofibroblasts, contributing to the heterogeneity and high plasticity of the tumour [27].

Firstly, an increase in the inflammatory response leads to proliferative inflammatory atrophy (PIA) (Figure 2c), which is characterized by a hyperproliferative response of the epithelia. Repeated cycles of cell injury and regeneration result in increased oxidative stress mediated by the inflammatory system [31]. These impact the increased levels of glutathione S-transferase P1 (GSTP1) in response to oxidant stress [32]. PIA regions also show an increased expression of p27Kip1 (also known as cyclin-dependent kinase inhibitor 1B or CDKN1B), which inhibits the cell cycle, and B-cell lymphoma-2 (Bcl-2), which regulates apoptosis [32]. However, there is downregulation of tumour suppressor genes, including the transcription factor NK3 homeobox 1 (*NKX3.1*), which is essential for maintaining prostate cell fate and suppressing PCa initiation, and the tumour suppressor phosphatase and tensin homolog (*PTEN*) gene [33]. PIA has been proposed as a precursor of prostatic intraepithelial neoplasia (PIN) and PCa (Figure 2d) [34]. The mechanisms underlying these transitions are not fully understood, but evidence suggest that PIA is an intermediate state to PIN [31]. This hypothesis is supported by the fact that PIA regions overlap with regions of tumour tissue, recognizing the sequence of events that occur after inflammation-induced prostate carcinogenesis [31].

PIN is characterized by luminal epithelial hyperplasia, reduction of basal cells, enlargement of nuclei, increased proliferative markers, loss of NKX3.1, and alteration of mitotic rates of the epithelial bilayer (Figure 2d) [35,36]. Studies have also demonstrated that PIN tissue overlaps with tumour tissue, supporting the hypothesis that PCa originates from this prostate state [37]. This phase of the disease is also characterized for telomere shortening [38], increase of genomic instability, chromosome mutations [39], and telomerase activation to restore telomere length, avoiding replicative cell senescence [39,40]. Studies also demonstrated the involvement of the ETS transcription factor rearrangements in PIN [41]. One of these mutations create a transmembrane serine protease isoform 2 (TMPRSS2)-ERG fusion gene, which increase the predisposition to tumour progression [42]. Additionally, missense mutations of the speckle-type POZ protein (SPOP) gene frequently occur in this phase. SPOP is a tumour suppressor protein and substrate adaptor of the cullin 3-RING-ubiquitin ligase (CUL3). SPOP mutations disrupt substrate binding and ubiquitination, leading to increased expression of oncogenic substrates [43].

Continued elevation of the inflammatory response and genetic alterations cause cellular damage leading to cancer progression. Increased influx of T cell infiltrates [44], tumour associated macrophages [45], and B cells [46] result in continuous damage to prostate tissue, leading to the production of a reactive milieu of pro-inflammatory cytokines and growth factors (Figure 2e). These events ultimately lead to the alteration of the epithelial niche toward a pro-inflammatory phenotype [47]. Concurrently, there is luminal cell hyperproliferation, loss of basal cells, and PTEN, and disintegration of the basement membrane, allowing tumour and tumour microenvironment cells to invade surrounding tissues [48]. During this phase, aberrantly differentiated cells acquire a telomerase-positive signature to maintain clonal heterogeneity and viability [49]. Additionally, there is a loss of the tumour suppressor retinoblastoma 1 (RB1) [50].

## 5. Current Therapeutic Strategies Used in Prostate Cancer

The current therapeutic strategies for PCa encompass a range of approaches, including active surveillance, surgery, radiation therapy, chemotherapy, hormonal therapy, and immunotherapy, often in combination [19].

Active surveillance involves a tailored plan for each patient, considering specific needs. Typically, this includes periodic PSA tests (once or twice a year), a DRE (once a year), and a prostate biopsy (every 1 to 3 years). This decision is made by a team of clinicians, generally for patients with lower-risk PCa, a life expectancy of 10 years or more, overall patient health, tumour characteristics, and potential side effects [51]. Illness uncertainty could be a potential adverse effect of this type of control.

Surgery aims to remove cancer and the type of procedure is determined by factors such as tumour size, location, and metastasis. Radical prostatectomy, which removes the entire prostate gland, is recommended for patients with local recurrence without metastasis after treatments such as radiotherapy, brachytherapy, or cryotherapy. Nevertheless, surgery is often associated with significant morbidity, including erectile dysfunction, urinary incontinence, and infertility [52].

Radiotherapy employs high-energy radiation, such as X-rays or gamma rays, to eliminate PCa cells. It can be used as an alternative to surgery or after surgery to prevent cancer recurrence. External beam radiation therapy delivers precise radiation to prostate tissue, sparing healthy cells, and is considered effective for intermediate-risk and high-risk PCa [53]. However, studies have shown that PCa cells can adapt to radiotherapy, increasing the risk of disease recurrence [54]. Side effects may include high urinary frequency, dysuria, diarrheal, proctitis, erectile dysfunction, and urinary incontinence [55]. Brachytherapy involves the direct delivery of radiation into the prostate gland using seeds, injections, or wires guided by transrectal ultrasounds. This technique can help preserve continence and erectile function but requires anaesthesia and may raise the risk of urinary retention [56].

Chemotherapy employs anticancer drugs to inhibit the survival, proliferation, and metastasis of tumour cells. Docetaxel is a common choice for PCa, acting by binding to β-tubulin and inhibiting microtubule depolymerization, mitotic cell division, and promoting apoptosis. Resistance to this drug may involve the upregulation of the multidrug resistance (MDR) 1 gene, which encodes P-glycoprotein [57]. Cabazitaxel is a second-generation therapy designed to counter docetaxel resistance, with low affinity for P-glycoprotein due to an additional methyl group [58]. Enzalutamide, a second-generation androgen receptor inhibitor, can act through competitive inhibition of androgen binding to the androgen receptor, inhibition of nuclear translocation, co-factor recruitment, and inhibition of DNA binding by the activated androgen receptor [59]. However, chemotherapy is associated with severe side effects in patients, including anaemia, neutropenia, nausea, vomiting, diarrheal, mucositis, ototoxicity, nephrotoxicity, pulmonary toxicity, and neurotoxicity [60].

Hormonal therapy, also known as androgen deprivation therapy, is commonly employed in advanced and metastasized PCa. It involves blocking hormone production, including testosterone, leading to the inhibition of androgen and androgen receptor signalling. This can be achieved through the use of luteinizing hormone-releasing hormone (LHRH) analogues or antagonists. LHRH analogues, like leuprolide, goserelin, triptorelin, and histrelin, initially increase luteinizing hormone (LH) and follicle-stimulating hormone (FSH) levels by stimulating pituitary receptors. Subsequently, these drugs downregulate pituitary receptors, resulting in reduced LH and FSH levels and subsequent testosterone inhibition. LHRH antagonists, on the other hand, block pituitary receptors, triggering testosterone inhibition [61]. However, this therapy is associated with side effects such as hyperlipidaemia, fatigue, hot flashes, a flare effect, osteoporosis, insulin resistance, cardiovascular disease, anaemia, and sexual dysfunction [62].

Immunotherapy offers a promising approach to PCa treatment by manipulating the immune system’s response to fight cancer cells. Sipuleucel-T was the first FDA-approved immunotherapy for PCa. It involves collecting a patient’s immune cells, specifically dendritic cells, exposing them to a PCa protein, and then reinfusing these activated cells into the patient. This treatment has shown potential in extending survival in some patients with advanced PCa [63]. Checkpoint inhibitors, which block proteins like PD-1 and PD-L1 that prevent immune cells from attacking cancer cells, have been successful in other cancer types but have shown limited efficacy in PCa [64]. Chimeric antigen receptor T-cell therapy (CAR-T) genetically engineers a patient’s T cells to target specific antigens on cancer cells, and it is being explored as a potential treatment for advanced PCa, focusing on antigens like prostate-specific membrane antigen [65]. Additionally, various vaccine-based approaches for PCa, including dendritic cell vaccines and viral vector-based vaccines, are under investigation to stimulate the patient’s immune system to recognize and combat PCa cells [66].

## 6. Inflammation and Prostate Cancer

Around 20% of all cancers are related with inflammation [67]. Different studies have suggested that inflammation plays a crucial role in prostatic carcinogenesis and tumour progression [68]. In fact, numbers have shown that inflammatory tissue is prevalent in 77.6% of prostate biopsy tissues and can even reach up to 80% in the general population [69].

Inflammation serves as a crucial immune response that occurs in the aftermath of injury or infection. It functions as an essential defence mechanism responsible for clearing pathogenic materials and debris from damaged tissues while also initiating the wound healing process [70]. Studies have demonstrated the role of neutrophils [71,72], B cells [73,74], T cells, [75,76,77], and macrophages [45,78,79,80] in PCa [81]. Persistent tissue damage leading to chronic inflammation or dysregulation of the inflammatory mechanisms promote increased release of inflammatory mediators, cytokines recruitment, expansion of leukocytes, and genomic instability [82]. Consequently, these processes can cause DNA damage in epithelial cells, which accumulates DNA mutations [83].

The impact of inflammation on PCa has been demonstrated in a population-based case-control trial [84]. This study revealed a 23% reduction in the risk of PCa associated with non-steroidal anti-inflammatory drugs (NSAIDs) and an even stronger association among patients treated with cyclooxygenase 2 (COX-2) inhibitors. Another study reported that daily aspirin consumption led to a long-term reduction of 29% in PCa risk compared to non-consumers [85]. Additionally, statins were found to be correlated with a reduced risk of PCa by inhibiting 3-hydroxy-3-methyl-glutaryl-coenzyme A (HMG-CoA) [86,87]. These results underscore the role of inflammation as a driver of prostate carcinogenesis.

### Origins of Inflammation

The initial cause of prostatic inflammation is difficult to predict. It can arise from the dysregulation of inflammatory pathways or by an external agent which drives inflammation. Environmental factors that have been identified as potential drivers of PCa include bacterial infections including sexual transmitted infections [88], viral infections [89], androgen and androgen receptor levels [90], diet and obesity [91], urine reflux [92], and genetic predisposition [93]. Bacterial and viral infections can exacerbate inflammation in the prostate, potentially leading to prostatitis. Notably, not all prostate infections progress to PCa, and the contribution of these infections to PCa remains unclear and inadequately covered in the literature, requiring further studies. The link between a high-fat diet and obesity as risk factors for PCa development remains controversial. This correlation has been explored due to variations in PCa incidence and mortality across different geographic and cultural regions [94]. Furthermore, evidence indicates that obesity, weight gain, and increased visceral fat are significantly associated with an elevated risk of biochemical recurrence after primary prostatectomy, more aggressive disease, and increased PCa-specific mortality [95]. Studies have also suggested that diets rich in red meat, charred meat, and saturated fats are risk factors for PCa [91]. Urine reflux has been proposed as a cause of chronic inflammation in the prostate due to chemical irritation resulting from the accumulation of uric acid. Several studies have demonstrated that uric acid is the primary chemical compound involved in this type of damage [96]. Notably, the hereditary risk of PCa is greater than that of any other human cancer [93]. Genome-wide association studies have identified genetic loci associated with PCa and emphasized the significance of family history in PCa development [97,98]. The diversity of genetic abnormalities identified suggest that there is no single dominant molecular pathway for prostatic carcinogenesis but rather a combination of alterations [35,99]. The exact mechanisms involved in inflammation driven PCa are not fully understood. Nevertheless, somatic genome alterations in genes such as ribonuclease L (*RNASEL*), macrophage scavenger receptor 1 (*MSR1*), macrophage inhibitory cytokine-1 (*MIC-1*), intercellular adhesion molecule (*ICAM*), and Toll-like receptors (*TLR*) are among the most well-identified factors [35]. Due to the complexity of the process, hundreds of genes are implicated in the inflammatory response that leads to PCa. Therefore, new technology platforms and approaches are urgently needed to screen, identify, and correlate genes involved in the entire pathway. These advances could be pivotal in predicting, early detecting, treating, and preventing PCa development and progression.

## 7. Role of Leukocytes in Prostate Cancer

Chronic inflammation is evident in malignant prostate tissue, with PCa samples exhibiting a higher percentage of T lymphocytes and macrophages compared to neutrophils, eosinophils, and B cells typically found in acute inflammatory responses [100]. A study conducted in the UK Biobank found no correlation between white blood cells, including neutrophils, eosinophils, basophils, monocytes, and lymphocytes, and PCa diagnosis. However, a higher total white blood cell count, and neutrophil count were associated with an increased risk of PCa-related mortality [101].

The use of the CIBERSORT method to examine the relative proportion of immune cell populations in PCa revealed infiltrated T cells, CD8^+^ T cells, resting memory CD4^+^ T cells, and total macrophages counted, respectively, 39%, 13%, 20%, and 13% [102]. An immunophenotypic analysis from isolated prostatectomy specimens demonstrated an increase of CD11b^+^CD68^+^CD14^+^HLA-DR^high^ monocytes and CD11b^+^CD68^−^CD16^+^HLA-DR^low^ monocytes among the CD11b^+^ myeloid cells, a high fraction of CD8^+^ T cells within total CD45^+^ immune cells in PCa tissues and an increase of CD4^+^ forkhead box subfamily 3^+^ (FOXP3^+^) in high-grade PCa compared to low-grade PCa [103].

The discrepancies observed in different studies may be attributed to the heightened heterogeneity of PCa, leading to variations in immune subset phenotypes depending on the tissue samples collected from each patient. Moreover, different immune populations may express distinct immunophenotypic markers and be programmed toward either a pro-tumoral or anti-tumoral phenotype (Table 2).

### 7.1. Neutrophils

Neutrophils, originating from hematopoietic stem cells, are among the first immune cells recruited after an insult. They possess a short lifespan to prevent excessive tissue damage, owing to their high plasticity and robust effector response [150]. When recruited to a damaged area, neutrophils release proteases, including neutrophil elastase, neutrophil extracellular traps (NETs), and reactive oxygen species (ROS), which exacerbate damage and contribute to the development of chronic inflammation [151]. Under normal circumstances, neutrophils can shift their function towards immunosuppression, thus regulating the production of pro-inflammatory mediators. However, in disease states, this shift may not occur correctly, leading to the development of carcinogenesis [152]. Therefore, neutrophils serve as a crucial link between inflammation and cancer. A study has observed a correlation between low neutrophil counts and a positive PCa biopsy, while elevated neutrophil counts may indicate a benign prostate biopsy [153]. These results can predict the progression from an acute response, characterized by increased neutrophil levels, to a carcinogenic phenotype dominated by chronic inflammation [154]. Tumour associated neutrophils (TANs) have been reported in cancer-affected regions. TANs, along with regular neutrophils, secrete substantial amounts of matrix metalloproteinase (MMP)-9, which play a role in the degradation of the extracellular matrix and cancer progression [104].

TANs are a complex population in the tumour microenvironment, associated with poor outcomes in some PCa studies [72] and demonstrating antitumoral effects in others [155]. In vitro assays showed that coculture of human PCa cells in the presence of neutrophils leads to a reduction of cell growth via caspase activation [156]. These findings suggest that, as tumours progress, neutrophil cytotoxicity diminishes, allowing PCa to avoid neutrophil cytotoxic effects. Studies have linked neutrophils as crucial cells in PCa prevention. In bone metastatic PCa, there is an increased formation of neutrophils and NETs to limit the spread of infection and control metastasis [156]. The role of different inflammatory mediators produced by neutrophils and its role in cancer progression is summarized on Table 2.

### 7.2. Basophils

Basophils constitute approximately 1% of circulating white blood cells and serve as protectors against allergens, pathogens, and parasites. In an inflammatory context, basophils can migrate to inflammatory regions and promote M2-like macrophage polarization, highlighting the disparity in function between circulating and resident basophils [157].

Elevated basophil and basophil-to-lymphocyte ratio were associated with a poor outcome in metastatic hormone sensitive PCa [158]. Epithelial-derived pro-inflammatory cytokines including interleukin (IL)-33, IL-18, granulocyte-macrophage colony-stimulating factor (GM-CSF), and growth factors including IL-3, IL-7, transforming growth factor-beta (TGF-β), vascular endothelial growth factor A (VEGF) promote activation of basophils [159]. Several studies demonstrated that activated basophils can secrete different cytokines involved in PCa including IL-4, which promotes tumour-promoting Th2 inflammation [112,160] and M2 macrophage polarization related to a poor prognosis [161], IL-13 [157], and tumour necrosis factor-alpha (TNF-α) [162]. Studies also suggested the role of basophils in angiogenesis. Basophils release high amount of VEGFA, a potent proangiogenic molecule [115]. Basophils are a source of hepatocyte growth factor (HGF), a powerful angiogenic factor in tumours [116]. Human basophils also express angiopoietins (ANGPT) 1 and ANGPT2 mRNAs which are involved in vascular permeability [117]. Other studies showed the protective role of basophils in cancer development [114]. Low levels of circulating basophils correlated with higher size and extend of the tumour, higher number of lymph nodes and poor survival in colorectal cancer patients [163]. The effects of different inflammatory molecules produced by basophils in cancer are described in Table 2.

While most data on the role of basophils in cancer progression pertains to cancers other than PCa, additional studies are needed to elucidate the mechanisms by which basophils influence PCa.

### 7.3. Eosinophils

Eosinophils constitute 1–4% of white blood cells and play a vital role in maintaining homeostasis and defending the host against infectious agents. They originate from multipotent CD34^+^ progenitors in the bone marrow [164]. Under normal conditions they are located in spleen, lymph nodes and thymus. When activated, they have the capacity to modulate the immune response, including the phenotype of T cells. [165]. The migration and recruitment of eosinophils to the tumour microenvironment are orchestrated by eotaxins, namely CC chemokine ligand (CCL)11, CCL24, CCL26, and CCL5 which activate the CCR3 receptor, highly expressed on eosinophils [166]. Eosinophils secrete cytotoxic granules including eosinophil cationic protein (ECP), major basic protein (MBP), eosinophil derived neurotoxin (EDN) and eosinophil peroxidase (EPO). Additionally, they release pro-inflammatory mediators such as IL-2, IL-4, IL-5, TGF-β, TNF-α, GM-CSF, and interferon-gamma (IFN-γ) [167]. IL-5 is a key mediator for eosinophil growth, differentiation, and activation [168]. Moreover, eosinophils express adhesion molecules CD11a/CD18, allowing them to interact with tumour cells, indicating their role in cancer progression [169]. Histological analysis of PCa samples revealed an increase in eosinophils compared to healthy controls in correlation with age and Gleason score [170].

On the other hand, activated eosinophils inhibited PCa cell growth through upregulation of E-cadherin, a metastasis suppressor molecule [171]. Evidence demonstrated that incubation of PCa cell lines with activated eosinophils inhibited cell growth [172]. Treatment of patients with metastatic castration-resistant PCa with Sipuleucel-T led to an increase in eosinophil counts, correlated with improved survival and enhanced maximal T-cell proliferation responses [120]. The role of different cytokines and chemokines produced by eosinophils are labelled in Table 2.

To advance the understanding of eosinophils in the tumour microenvironment and their interactions with other immune cells, it is crucial to improve the technological detection of eosinophils and discover novel biomarkers for defining eosinophil subpopulations. This will provide insights into their ability to modulate various cells in different cancers, including PCa, and their role in cancer progression.

### 7.4. Mast Cells

Mast cells derive from CD34^+^/CD117^+^ hematopoietic stem cells in the bone marrow and they undergo maturation within target tissues [173]. Besides KIT activation, which is essential for mast cell development, several cytokines, including IL-3, IL-4, IL-9, IL-10, IL-33, and TGF-β, influence their growth and survival [174]. Mast cells exhibit significant plasticity and can adopt various phenotypes depending on the host’s genetic background and local or systemic factors [175]. These cells are characterized by the presence of numerous granules rich in histamine and heparin. Upon activation, mast cells can degranulate and release inflammatory mediators to combat pathogens [176]. This response leads to the synthesis of specific cytokines, including anti-inflammatory TGF-β, IL-10, as well as the proinflammatory associated cytokines IL-4, IL-6, and IFN-γ [131]. Evidence demonstrated that mast cells are present in several tumours [177,178]. Zadvornyi et al. [179] demonstrated that increased mast cell infiltration and degranulation were associated with malignancy of PCa. Another study demonstrated the potential of mast cells to promote PCa cell proliferation and epithelial mesenchymal transition which is linked with invasion and metastasis [180]. A study dissected that infiltrating mast cells in PCa suppress androgen receptor-MMP signalling promoting PCa cell invasion [181]. Intratumoral mast cell tryptase^+^/chymase^+^/CD117^+^ phenotype was founded in malignant PCa samples [182]. Additionally, a high extratumoral mast cell count was linked with a high risk of biochemical recurrence and PCa metastasis [183].

Release of IL-1, IL-4 and IL-6 from mast cells was associated with elimination of tumour cells and rejection of tumours [129]. Other studies demonstrated that IL-1 is linked with tumour growth, angiogenesis, macrophage recruitment and metastasis [128]. Another study in human PCa samples associated higher mast cell infiltrates with a better PCa prognosis [184]. Moreover, mast cells can contribute to angiogenesis inhibition through secretion of prostaglandin D2 (PGD2) [132]. Table 2 dissects the role of each cytokine released by mast cells on cancer progression.

### 7.5. Macrophages

Macrophages are vital phagocytic cells integral to the innate immune response. The primary sources of macrophages are monocytes, which circulate in the blood, and tissue-resident macrophages originating from the yolk sac. These cells are recruited and activated by the specific microenvironment in which they operate. In the context of the tumour microenvironment, macrophage activation plays a significant role in influencing tumour development, progression, metastasis, immune regulation, and angiogenesis [79].

Activated macrophages can be classified into two main categories: M1-like macrophages, which promote inflammation to combat pathogen invasion and cancer, and M2-like macrophages, which are associated with tissue repair and support tumour progression [185]. M1-like macrophages secrete proinflammatory mediators including IL-12, TNF-α, chemokine (C-X-C motif) ligand (CXCL)-10, IFN-γ, and nitric oxide synthase (NOS), whereas M2-like macrophages produce anti-inflammatory IL-10, IL-13, and IL-4, arginase-1, the mannose receptor CD206, and scavenger receptors [186,187]. The polarization of macrophages into M1 or M2 phenotypes depends on the signals present in the microenvironment.

Research has demonstrated that tumour-associated macrophages (TAMs) often acquire a tumour-suppressive M2-like phenotype, contributing to the development of carcinogenesis [188]. The release of IL-1β, IL-8, TNF-α, TGF-β [133,134,135], MMP-2, and MMP-9 [136] by macrophages is involved in epithelial-mesenchymal transition (EMT), which promotes cancer cell invasion and metastasis. It is believed that metastatic processes could not be a late event in tumour progression. The primary tumours could prime the metastatic organ before tumour cell arrival. Macrophages are involved in the formation of this pre-metastatic niches. They are mobilized to bloodstream and are then clustered in these regions by CCL2, CSF-1, VEGF, platelet-derived growth factor (PDGF), TNF-α, and TGF-β [189,190]. The role of inflammatory mediators produced by macrophages in cancer is discussed in Table 2.

Given their ability to modulate the tumour microenvironment, the strategic targeting of macrophages has emerged as a promising approach in the development of new strategies for treating PCa. This approach holds the potential to provide novel and effective therapeutic alternatives for PCa patients.

### 7.6. T Cells

T cells play a crucial function in the adaptive immune response and were identified as key cells in the PCa tumour microenvironment [191]. These cells originate from the bone marrow and comprise various subtypes, including CD8^+^ T cells, CD4^+^ T cells, Th17 and regulatory T cells (Tregs). CD8^+^ T cells, known as cytotoxic T cells, exert their effects directly on infected cells. They predominantly secrete immune mediators such as IFN-γ, TNF-α, IL-2, granzyme, and perforin. In contrast, CD4^+^ T cells, or T helper cells, orchestrate immune responses by activating B cells and CD8^+^ T cells. They can be further categorized into Th1, Th2, Th17, and regulatory T cell (Treg) subsets. Th1 cells release proinflammatory cytokines, including IFN-γ, TNF-α, and IL-2, while Th2 cells secrete IL-4, IL-5, IL-13, IL-25, and IL-10, driving an anti-inflammatory response [192]. Th17 cells, characterized by secretion of IL-17A, IL-17F, IL-21, and IL-22, have been implicated in PCa metastasis. Studies have shown that the loss of Th17 function can hinder the development of microinvasive PCa in murine models [141]. Tregs were firstly defined as CD4^+^CD25^high^ cells and were found to be increased in PCa patients [193]. These cells play a vital role in discriminating self from foreign antigens and can either activate or suppress immune responses. Tregs release immunosuppressive cytokines, including IL-10, TGF-β, IL-35, CD39, CD73, and indoleamine 2,3-dioxygenase (IDO) [194]. The immunosuppressive functions of Tregs favour tumour progression, and elevated Treg levels in PCa patients have been associated with poorer survival outcomes [80]. CD8^+^ T cells have been associated with a good prognosis and a study identified CD8^+^CD44^+^ population as important for reducing the tumour burden [195]. On the other hand, accumulation of CD4^+^ T cells in the PCa tumour microenvironment was associated with a poor survival [75]. In fact, this population is increased in PCa patients in comparison to healthy controls. An increase of CD4^+^ T cells was also associated with increased chemo-resistance to docetaxel in PCa cells [76]. Later, Kaur et al. [196] showed an association between increased transcription factor FOXP3^+^ Treg cells and risk of metastasis. Another study identified CD4^+^FOXP3^+^ Tregs and CD8^+^FOXP3^+^ Tregs increased in PCa samples and associated with increased risk of death [77,197].

Understanding the intricate roles of T cells, including their subtypes and cytokine profiles (Table 2), is essential for deciphering the complex immune landscape within the PCa microenvironment.

### 7.7. B Cells

B cells originate in the bone marrow and have the ability to migrate to the spleen and lymph nodes. Naïve B cells undergo activation into plasma cells in response to specific antigens during their development, leading to proliferation and differentiation. The maturation of B cells results in changes to their epitopes, and their characterization relies on CD markers such as CD19, CD20, CD21, CD40, and CD79b [198]. B cells can influence the tumour microenvironment through various mechanisms, including antibody presentation, antibody production, and cytokine secretion [199]. Studies have demonstrated that B cells activate CD4^+^ T cells, resulting in the accumulation of T cells in the tumour microenvironment and the differentiation of CD4^+^ and CD8^+^ T cells into distinct phenotypes [200]. Interactions between CD20^+^ B cells and T cells in the tumour microenvironment have been shown to impact the protective function of T cells [201]. Regulatory B cells (Bregs) are frequently found in advanced hepatocellular, gastric, and PCa, suggesting their potential influence on tumour development and progression [73,202,203]. Bregs are associated with an anti-immune function, as they release immunosuppressive molecules such as IL-10, IL-35, and TGF-β, which hinder the activity of T cells [146,204,205,206].

Literature data suggest that the infiltration of B cells increases the risk of adverse events in prostate carcinogenesis and malignancies. The effect of inflammatory modulators produced by B cells on tumour progression is described in Table 2.

### 7.8. Overall Remarks

Overall, IL-1, IL-6, IL-2, IL-4, IL-7, IL-8, IL-10, IL-17, IL-23, TNF-α, TGF-β, IFN-γ, VEGF, and GM-CSF are the main inflammatory mediators involved in PCa (Figure 3).

IL-1 and IL-6 promote cancer growth, proliferation, and progression [207,208]. IL-1 is increased in PCa and induces immunosuppressive function of mesenchymal stem cells [209,210]. IL-6 is increased in PCa, induces EMT and metastasis, increases the expression of androgen receptor, and induces infiltration of T cells into the tumour microenvironment [211,212]. IL-2 has been found to stimulate Tregs, with some studies associating it with tumour growth and progression, while others suggest its potential anti-tumour activity [213]. IL-4 increases the expression of androgens, activates the JNK pathway, and promotes tumour progression [214]. IL-7 induces EMT and cancer metastasis [215]. IL-8 stimulates proliferation of prostate stromal cells, regulates the expression of MMPs, promotes PCa progression, angiogenesis, and metastasis [216]. IL-10 inhibits anti-tumour responses and regulates androgen signalling, promoting cancer metastasis [217]. IL-17 promotes PCa growth, angiogenesis, and metastasis [218], increases the expression of programmed death-ligand 1 (PD-L1) and COX-2 and induces the release of IL-6 and IL-8 [219]. IL-23 regulates the androgen response and Th17 survival [217]. TNF-α and TGF-β are able to promote PCa progression and metastasis [220]. TNF-α upregulates the expression of PD-L1, and its control is indicative of tumour cell behaviour [219,221]. TGF-β induces EMT, inhibition of anti-tumour activity, reduces the expression of major histocompatibility complex (MHC)-I, regulates angiogenesis, the formation of the premetastatic niche, and metastasis in bone [222,223,224]. IFN-γ induces the release of IL-6 and IL-8 and promotes anti-tumour response [225]. VEGF contributes to angiogenesis, formation of premetastatic niche, tumour microenvironment remodelling, tumour invasion, and metastasis [224,226]. GM-CSF stimulates leukocytes and increases tumour antigen presentation to effector T cells [227,228].

The activity of interleukins is primarily modulated through the Janus Kinase/signal transducers and activators of transcription (JAK/STAT) pathway. This signalling pathway is integral to normal development, cellular homeostasis, cell proliferation, differentiation, and apoptosis [229]. Ligand binding initiates the multimerization of receptor subunits, leading to the activation of the JAK/STAT pathway and the transmission of signals through the phosphorylation of receptor-associated JAK tyrosine kinases. Consequently, activated JAKs induce the phosphorylation and activation of STATs. This phosphorylation prompts the dimerization of STATs via their conserved SH2 domain, subsequently allowing them to enter the nucleus. Within the nucleus, STATs bind to specific DNA sequences, either stimulating or suppressing the transcription of target genes [230]. It was reported that JAK/STAT3 inhibition suppress PCa cell growth and increases apoptosis [231]. BRCA1 via JAK1/2 and STAT3 phosphorylation can induce cell proliferation and inhibit cancer cell death [232]. The androgen receptor could also activate JAK/STAT3 and stimulate cell proliferation and antiapoptotic effect increasing tumour invasion [233,234].

NF-κB is a transcription factor predominantly activated by cytokines such as TNF-α in PCa. In androgen-dependent PCa, IL-6 and VEGF stimulates the increase of the expression of NF-κB [235]. NF-κB targets a transcription regulatory element of PSA and correlates with cancer progression, chemoresistance, and PSA recurrence [236].

Growth factors including VEGF, epidermal growth factor (EGF), insulin-like growth factor (IGF)-1, HGF, and TGF-β are key players in the receptor tyrosine kinase (RTK) signalling pathway. These growth factors activate the extracellular signal-regulated kinases (ERK)/MAPK or PI3K/AKT/mTOR mechanisms [237]. Growth factor receptors possess RTK activity, and their binding to ligands leads to the activation of transcription factors, resulting in the altered expression of genes associated with cell growth, proliferation, and survival [238]. IGF-1 functions as a positive growth-promoting signal transduction pathway, while FGF plays a dual role as a positive growth factor and an angiogenic growth factor. On the other hand, TGF-β serves as a negative growth factor, regulating cell differentiation and proliferation [239]. Studies demonstrated that alterations on the expression of TGF-β, EGF and their receptors correlates with PCa progression and biochemical recurrence [240,241]. The phosphoinositide 3-kinase (PI3K)/protein kinase B (AKT) pathway is often upregulated due to the loss of the tumour suppressor PTEN, which negatively regulates the PI3K/AKT pathway [242]. It has been demonstrated that the aberrant PI3K/AKT pathway disturbs the action of ERKs, thereby supporting androgen receptor-independent growth in PCa [243]. Overexpression of growth factors promotes the activation of Ras and MAPK pathways [244]. Upon activation, MAPKs phosphorylate transcription factors such as c-Jun, c-Fos, ATF2, and p53. Additionally, ERK or p38 MAPKs can activate MAPK interacting protein kinases 1 and 2 (MNK1 and MNK2), which controls signals involved in mRNA translation [245]. Interestingly, MNKs have been found to be overexpressed in PCa [246].

As discussed in this review, inflammatory signalling plays a significant role in the development and progression of PCa. Considering these findings, therapeutic strategies targeting inflammatory signalling pathways in PCa may help manage the disease and potentially improve outcomes.

Ongoing research is exploring new treatments and strategies, especially those utilizing natural bioactive compounds, to mitigate the severe side effects, radiotherapy resistance, and recurrence of PCa.

## 8. Polyphenol Compounds in Prostate Cancer

In recent years, the utilization of natural compounds in cancer treatment has gained substantial attention and research interest for several compelling reasons. These compounds, when administered in appropriate doses and forms, often exhibit fewer adverse effects in comparison to conventional cancer treatments, thereby enhancing the overall quality of life for cancer patients. Furthermore, specific natural compounds can target distinct signalling pathways and molecular processes involved in cancer growth and progression. They can be utilized in conjunction with chemotherapy or radiation therapy to augment the delivery of therapeutic drugs to cancer cells, thereby improving their effectiveness. Additionally, these natural compounds, primarily polyphenols, possess antioxidant and anti-inflammatory properties that are of utmost importance in cancer treatment. Notably, they can also bolster the body’s natural defences, facilitating a more effective targeting and elimination of cancer cells [247].

Polyphenols are plant secondary metabolites that have garnered significant attention in cancer research owing to their potent antioxidant capabilities in neutralizing free radicals [248]. These polyphenols fall into various categories, including flavonoids, phenolic acids, lignans, and stilbenes [7]. They typically feature one or more hydroxyl groups attached to the *ortho*, *meta*, or *para* positions on a benzene ring. These hydroxyl groups are highly reactive, readily donating electrons or hydrogens to neutralize free radicals, thus playing a crucial role in their antioxidant activity. The aromatic rings in phenolic compounds form conjugated systems, enabling the delocalization of electrons, which contributes to their stability (Figure 4). This structural characteristic enhances their ability to scavenge free radicals and prevent their propagation within cells.

The effectiveness of each polyphenol relies on its specific chemical structure, with flavonoids standing out for their significant anti-cancer properties. The architecture of flavonoids features two benzene rings connected by a heterocyclic pyran ring, providing structural stability, and facilitating electron delocalization, which in turn enhances their antioxidant potential. Moreover, the presence of a double bond between C2 and C3, a hydroxyl group in *ortho*-positions, carbonyl conjugation at C4, and methoxy groups on the benzene rings greatly enhances their ability to donate electrons, effectively neutralizing and stabilizing free radicals [249]. In addition to these attributes, certain polyphenols can also exhibit chelating properties, binding to metal ions that can otherwise trigger the production of free radicals. This chelation process serves to prevent the formation of reactive species, further bolstering their antioxidative impact [9].

The role of polyphenol compounds in PCa was demonstrated by several authors (Table 3).

Curcumin is the active compound in turmeric, a spice commonly used in Indian cuisine. It promoted apoptosis and inhibited angiogenesis and metastasis in DU-145 PCa cells [250] and inhibited cell proliferation, migration, and invasion in PC-3 and DU-145 cells by regulating the miR-30a-5p/PCLAF axis [251]. In another study, curcumin was found to block the growth of LNCaP xenografts through several mechanisms. It induced apoptosis, hindered proliferation, and upregulated the expression of key factors, including TRAIL-R1/DR4, TRAIL-R2/DR5, Bax, Bak, p21/WAF1, and p27/KIP1. Simultaneously, it curbed the activation of NF-kB and its downstream gene products, including cyclin D1, VEGF, uPA, MMP-2, MMP-9, Bcl-2, and Bcl-xL [252]. Curcumin analogues inhibited growth and progression of PC-3-induced tumours in vivo [253,254].

Anacardic acid is the active phenolic lipid found in the *Amphipterygium adstringens* plant. It was reported that this phenolic acid inhibited PCa angiogenesis by targeting the proto-oncogene tyrosine-protein kinase (Src)/focal adhesion kinase (FAK)/rhodopsin (Rho) guanosine triphosphate (GTP)ase signalling pathway [255]. It mediates PCa by inhibiting cell proliferation and inducing G1/S cell cycle arrest and apoptosis [256].

Caffeic acid is a hydroxy-cinnamate metabolites existent in plant tissues. Caffeic acid derivates were involved in anti-proliferative effects by alterations in oestrogen receptors (ER)-α and ER-β abundance [257]. Other studies showed that caffeic acid-phenyl ester (CAPE) treatment suppressed proliferation and cell cycle progression in PC-3 cells [258].

Ellagic acid is a polyphenolic compound present in fruits and berries. Studies revealed that it modulates apoptosis inducing factor (AIF), leading to an increase in ROS levels and caspase-3 and a reduction in TGF-β and IL-6 [259]. Another study showed that ellagic acid inhibited invasion and motility of PCa cells [260].

Gallic acid is ubiquitously present either in free form or, more commonly, as a constituent of tannins in red and white wines. Gallic acid inhibited cell viability in DU-145 and 22Rν1 PCa cells by promoting apoptosis [261] and inhibited tumour growth in DU-145 and 22Rν1 PCa xenografts [262].

Resveratrol is one of the best studied stilbenes and is found in grapes. Resveratrol inhibited cancer cell growth, promoted cell cycle arrest and apoptosis in PCa cells [263,264,265] and, interestingly, increased the sensitivity of PCa cells to ionizing radiation [266]. Piceatannol is a metabolite bio transformed from resveratrol also with impact in PCa. Studies suggested that the effect of resveratrol in PCa cells is partially explained through its conversion to piceatannol [290]. This compound inhibited migration by a decrease in IL-6/STAT-3 signalling [267], and delayed G1 cell cycle progression by inhibition of CDK2 and CDK4 [268] in DU-145 PCa cells. Other studies showed that piceatannol inhibited TNF-α-induced invasion by suppression of MMP-9 activation via AKT-mediated NF-κB pathways in DU-145 PCa cells [269].

Pterostilbene is an antioxidant mainly found in berries and grapes. Its conjugate pterostilbene-isothiocyanate repressed proliferation, induced apoptosis by modulating PI3K/AKT and ERK/MAPK pathways, and down regulated androgen receptor expression in LNCaP cells [270]. This conjugate promoted cell cycle arrest in LNCaP cells by increasing p53 and p21 expression, which protects against the effects of AMPK activation [271].

The role of flavonoids including epigallocathechin-3-gallate (EGCG), fisetin, quercetin, apigenin, and proanthocyanidins in PCa was also revealed. EGCG is a polyphenol found in green tea. Studies showed that green tea polyphenols decreased risk and slower progression PCa [291] by modulating NF-κB/MAPK/IGFR/COX-2 signalling pathways, inhibiting protein kinases, and suppressing the activation of transcription factors [8]. EGCG promoted apoptosis via expression of caspase-9a [272], and suppressed pro-inflammatory cytokines, MMPs-2 and -9 in PCa cells [273].

Fisetin belongs to the flavanol subgroup of flavonoids. Studies showed that fisetin decreased the viability of LNCaP, 22Rν1, and PC-3 cells [274,275], suppressed cell proliferation by hypophosphorylation of eukaryotic translation initiation factor 4E-binding protein-1, and induced autophagic cell death in PCa cells through suppression of mTORC1 and mTORC2 complexes [276].

Quercetin is a plant pigment flavanol found in citrus fruits. Studies showed that quercetin decreased ROS, and increased superoxide dismutase (SOD) and catalase (CAT) in Sprague Dawley rats [277] but increased ROS production in DU-145 cell line acting as a pro-oxidant agent [278]. Quercetin decreased the expression of androgen receptor in LNCaP cells [279], decreased CDK2, cyclin E and D levels, VEGF, and promoted G0/G1 cycle arrest in PC-3 cells [280,281].

Apigenin is a naturally occurring plant flavone present in common fruits and vegetables. Apigenin mediated growth inhibitory responses through inhibition of histone deacetylases (HDACs) [282]. Apigenin induced PCa cell apoptosis via upregulation of p21 and subsequent inhibition of polo-like kinase (PLK)-1 transcription [283] and inhibition of class 1 HDACs and HDAC1 protein expression increasing the acetylation of Ku70 and the dissociation of Bax [284]. This molecule attenuated IGF-1/IGF binding protein-3 signalling associated with inhibition of p-AKT and p-ERK1/2, suppressing invasion and progression of PCa [285]. A study also showed that apigenin inhibit PCa progression via targeting PI3K/AKT/forkhead box FoxO pathways [286].

Proanthocyanidins, commonly known as condensed tannins, are found abundantly in various plants and foods. A study showed that proanthocyanidins downregulated MMP activity, upregulated endogenous tissue inhibitors of MMP’s (TIMP) activity in DU-145 cells [287] and affected the growth of androgen-dependent growth of PCa cells [288]. Proanthocyanidins also mediated inhibition of CDKs, cyclins, activation of tumour suppressors p21 and p27, Bcl-2/Bax ratio favouring apoptosis and induced cellular differentiation by increasing MAPK p44/42 in PCa cells [289].

It is evident that polyphenols play a crucial role in inhibiting the progression of PCa. However, the low bioavailability and limited absorption of free polyphenols in the human body have sparked interest in innovative strategies to enhance their delivery to target cells.

### Polyphenol-Gold Based Nanoparticles

In recent years, nanoparticles (NPs) have emerged as a promising approach to improve the delivery and targeting potential of polyphenols [292]. The encapsulation of polyphenols within NPs not only shields them from degradation but also enhances their solubility, thus improving absorption in the gastrointestinal tract. Moreover, the size, shape, and morphology of NPs can be precisely tailored to optimize polyphenol delivery and biocompatibility. Additionally, NPs can be functionalized with ligands to facilitate targeted delivery to specific organs and cells [293]. However, it is essential to note that conventional methods of AuNP synthesis are environmentally damaging, resource-intensive, and energy-consuming, with NPs prone to aggregation and toxicity within the human body [294]. This underscores the urgency of transitioning to environmentally-friendly, biocompatible, cost-effective, and sustainable green NP synthesis approaches [295]. The interest in developing polyphenol-based nanoparticles has been steadily growing. Polyphenols exhibit a high reducing capacity, allowing them to reduce metal ions and produce metal NPs. Their strong biocompatibility reduces NP toxicity, and they can serve as capping agents to prevent NP aggregation and enhance stability [296,297]. Furthermore, polyphenols exhibit synergistic effects with conventional drugs, potentially reducing the required drug concentrations and their associated toxicity. Additionally, the antioxidant properties of polyphenols can contribute to the overall efficacy of cancer treatment [298].

A comprehensive examination of polyphenol-based nanoparticles has been conducted to elucidate their impact on PCa progression (Table 4).

Curcumin incorporation in nanoparticles demonstrated high PCa cellular uptake [299]. Moreover, conjugation of this polyphenol with nanoparticles decreased proliferation and viability of PCa cells [300,301]. A combination of resveratrol and docetaxel downregulated the expression of NF-kB p65, COX-2, and upregulated cleaved caspase-3 [302]. Moreover, resveratrol improves internalization of NPs into PCa cells [303], increased their anti-proliferative activity, promotes cell cycle arrest. and decreased tumour cell viability [304,305]. EGCG nanoparticles had proapoptotic and antiangiogenetic effects on 22Rr1 cells [306], inhibited tumour growth in PC-3 cells [307], induced apoptosis and reduced viability of DU-145 cells [308], and inhibited tumour growth and secretion of PSA by increase of Bax, induction of poly (ADP-ribose) polymerases cleavage, and activation of caspases and apoptosis [309,310].

In conclusion, the encapsulation of polyphenols and their utilization in green NP synthesis represents an innovative therapeutic approach for PCa. This approach shows promise in attenuating the severe side effects and resistance associated with conventional treatments.

## 9. Conclusions

Nanotechnology has served as the foundation for remarkable industrial applications and exponential growth. Notably, in the pharmaceutical sector, nanotechnology has made a substantial impact on medical devices, including diagnostic biosensors, imaging probe delivery systems, and pharmaceuticals.

AuNPs offer diverse potential applications across various domains. Several in vitro studies demonstrated the anticancer potential of AuNPs as an anti-cancer agent as well as their stability, low toxicity, and specificity to PCa cells [6]. The traditional synthesis of AuNPs is highly polluting, wastes high levels of resources and energy, and nanoparticles can be toxic to the human body. A novel approach in the green synthesis of metal nanoparticles involves the use of bioactive natural compounds, rather than whole or partial plant extracts. Recent research has explored the synthesis and biomedical application of metal nanostructures based on phytochemicals. Natural phenolic acids offer a wide variety of metal ion bioreduction capabilities, making them ideal for creating biocompatible metal nanoparticles with numerous biomedical applications [296,297]. However, most metal nanoparticles produced with phenolic acids have only been applied in a limited range of biomedical applications. This is due to insufficient compelling results regarding biomedical properties and safety concerns, as compared to commonly chemically-coated metal nanoparticles. Comprehensive laboratory analysis considering various parameters such as size, shape, surface chemistry, the type of phenolic acids, and metal nanoparticles, must be conducted through rigorous animal models and well-designed molecular studies.

Recent reviews and meta-analyses have established a strong correlation between a history of clinical chronic prostatitis and the development of PCa in the general population [31]. The causes of prostate inflammation are multifaceted, ranging from bacterial triggers of prostatitis and sexually transmitted diseases to imbalances in oestrogen hormone levels, physical trauma, urine reflux into the prostate gland, and environmental factors such as diet [68]. Although prostate biopsy remains the gold standard for diagnosing prostate inflammation, various parameters, including laboratory biomarkers (cytokines) and clinical factors (familiar historical, age, prostatic calcifications, symptom severity, and response to therapy), can be valuable in everyday clinical practice when prostate inflammation is suspected [19]. Figure 3 illustrates the impact of prostatic inflammation and inflammatory mediators on tumour initiation, growth, and progression.

To develop more effective drug combinations and minimize toxicity, comprehensive studies are necessary to determine the optimal dosage of each drug within a combination and to monitor pharmacodynamic endpoints.

## Figures and Tables

**Figure 1 biomedicines-11-03140-f001:**
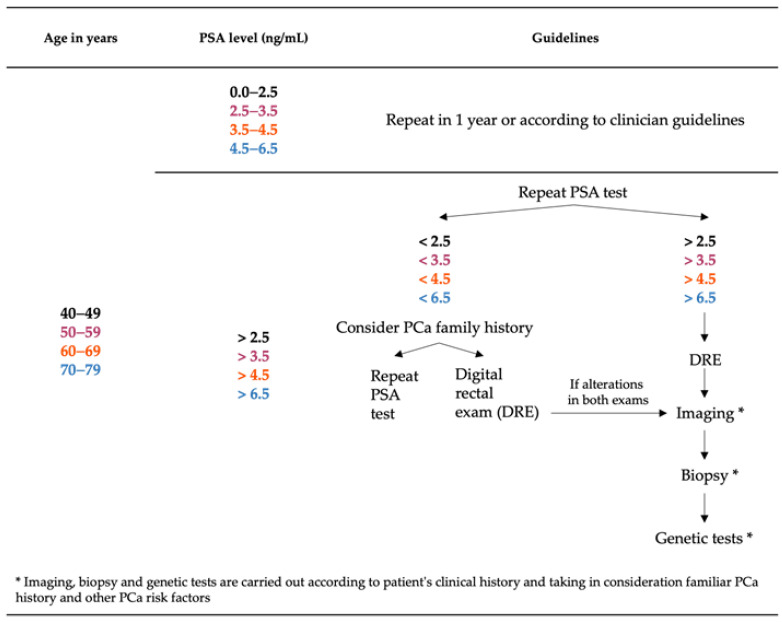
Diagnostic methods of prostate cancer (PCa) taking in consideration prostate-specific antigen (PSA) levels by age. The information represented in black, pink, orange, and blue correspond to patients between 40–49, 50–59, 60–69 and 70–79 aging, respectively.

**Figure 2 biomedicines-11-03140-f002:**
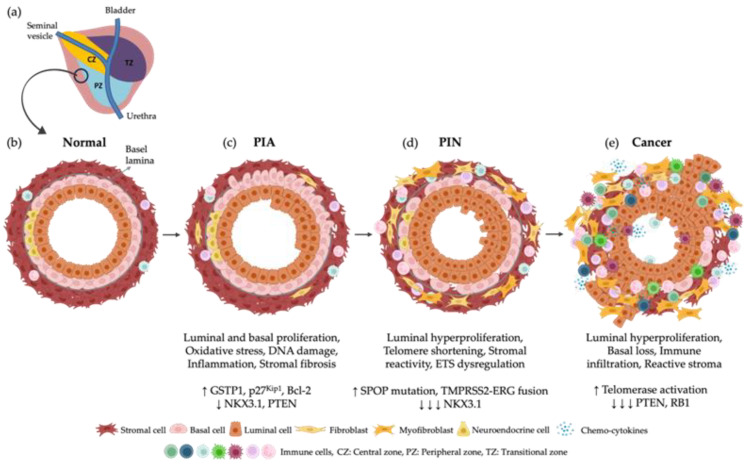
Pathophysiological mechanisms of prostate cancer (PCa). (**a**) Physiology of the human prostate; (**b**) Normal human prostate; (**c**) External or internal events can trigger an inflammatory response leading to proliferative inflammatory atrophy (PIA). PIA is characterized by hyperproliferation of epithelial cells, accompanied by an increase in glutathione S-transferase P1 (GSTP1), p27Kip1, and B-cell lymphoma-2 (Bcl-2), and a decrease in NK3 homeobox 1 (NKX3.1) and phosphatase and tensin homolog (PTEN) levels; (**d**) The subsequent phase of the disease, known as prostatic intraepithelial neoplasia (PIN), is marked by hyperproliferation of luminal cells, telomere shortening, activation and differentiation of fibroblasts into myofibroblasts, dysregulation of ETS transcription factor, speckle-type PO2 protein (SPOP) mutations, the presence of the transmembrane serine protease isoform 2 (TMPRSS2)-ERG fusion gene and loss of NKX3.1; (**e**) These events ultimately culminate in PCa, characterized by the loss of basal cells, activation of a pro-inflammatory phenotype, activation of myofibroblasts towards a pro-fibrotic state, telomerase activation and loss of PTEN and retinoblastoma 1 (RB1). Bcl-2: B-cell lymphoma-2, GSTP1: Glutathione S-transferase P1, NKX3.1: NK3 homeobox 1, PCa: Prostate cancer, PIA: Proliferative inflammatory atrophy, PIN: Prostatic intraepithelial neoplasia, PTEN: Phosphatase and tensin homolog, RB1: Retinoblastoma 1, SPOP: Speckle-type PO2 protein, TMPRSS2: Transmembrane serine protease isoform 2. Adapted from Packer et al. (2016) [1].

**Figure 3 biomedicines-11-03140-f003:**
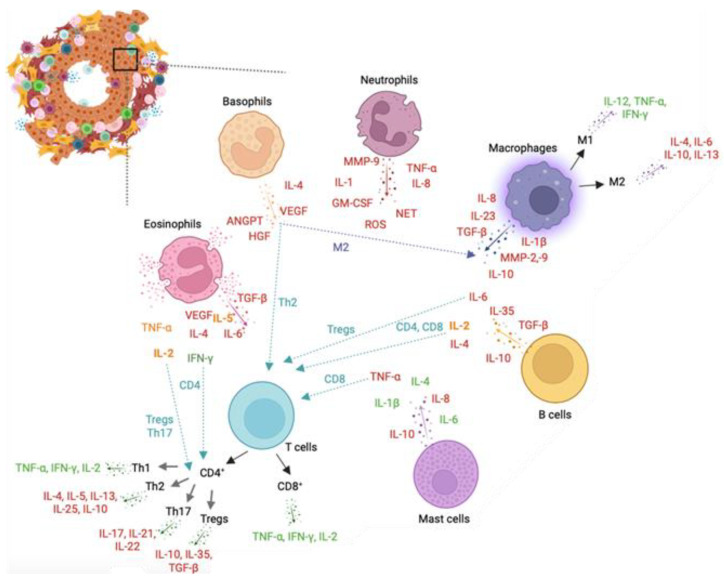
Leukocytes and leukocyte-mediated cytokines involved in prostate cancer (PCa). Representation of key cells in the prostate tumour microenvironment. Neutrophils mainly release matrix metalloproteinase (MMP)-9, interleukin (IL)-1, granulocyte-macrophage colony-stimulating factor (GM-CSF), reactive oxygen species (ROS), neutrophil extracellular traps (NETs), IL-8, and tumour necrosis factor (TNF)-α, all promoting PCa progression. Basophils promote activation of the M2-like phenotype in macrophages and Th2 response in T cells and release pro-tumour IL-4, VEGF, ANGPTs, and HGF cytokines. Eosinophils secrete pro-tumoral transforming growth factor (TGF)-β, IL-6, vascular endothelial growth factor (VEGF), and IL-4 cytokines and anti-tumoral interferon (IFN)-γ cytokine. Release of TNF-α, IL-2, and IL-5 from eosinophils can modulate a pro-tumoral or anti-tumoral activity depending on cell signals. IL-2 and IFN-γ stimulates activation of, respectively, Tregs/Th17 and CD4^+^ T cells. Mast cells secrete pro-tumoral TNF-α, IL-8, and IL-10 and anti-tumoral IL-1β, IL-4, and IL-6 cytokines. Tumour-associated macrophages secrete IL-1β, IL-8, IL-10, IL-23, MMP-2, MMP-9, and TGF-β, impacting tumour progression. Macrophages can shift to pro-inflammatory M1-like or anti-inflammatory M2-like phenotypes, influencing tumour outcomes. M1-like macrophages release IL-12, TNF-α, and IFN-γ, while M2-like macrophages secrete IL-4, IL-6, IL-10, and IL-13. T cells encompass CD4^+^ T cells (Th1, Th2, Th17, and Tregs) and CD8^+^ T cells, with Th1 having a pro-inflammatory response. Th2, Th17, and Tregs contribute to tumour progression. CD8^+^ T cells secrete IFN-γ, TNF-α, and IL-2, associated with a favourable prognosis in PCa. B cells release pro-tumoral IL-4, IL-6, IL-10, and TGF-β cytokines and the intermediate IL-2 cytokine. IL-2 and IL-4 stimulate CD4 and CD8 responses, while IL-6 activates Tregs in PCa. Green-coloured cytokines support tumour resolution and a positive prognosis. Red-coloured cytokines promote tumour growth, proliferation, and metastasis. Orange-coloured cytokines can trigger a pro- or anti-tumoral response in PCa. ANGPT: Angiopoietin, GM-CSF: Granulocyte-macrophage colony-stimulating factor, HGF: Hepatocyte growth factor, IFN: Interferon, IL- Interleukin, MMP: Matrix metalloproteinase, NETs: Neutrophil extracellular traps, PCa: Prostate cancer, ROS: Reactive oxygen species, TGF: Transforming growth factor, TNF: Tumour necrosis factor, Tregs: Regulatory T cells, VEGF: Vascular endothelial growth factor.

**Figure 4 biomedicines-11-03140-f004:**
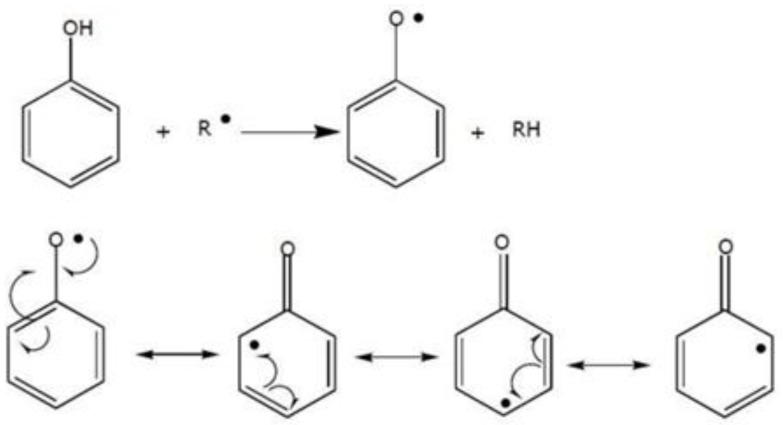
Chemical structure of phenols and their resonance states’ stability, imparting antioxidant activity for scavenging free radicals and inhibiting their propagation.

**Table 1 biomedicines-11-03140-t001:** Histological characterization of prostate cancer biopsies by Gleason Score. Adapted from the NCCN Guidelines for patients [19], Kweldam et al. (2019) [24], Ihamura et al. (2018) [25], and Avenel et al. (2019) [26].

Gleason Pattern	Gleason Sore	Grade Group	Risk	Histological Definition	Histological Image	Prognosis
3 + 3	6	1	Low	Individual, discrete, well-formed glands	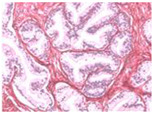	Low-grade Cancer
3 + 4	7	2	Low to intermediate	Well-formed glands with a few poorly-formed/fused/cribriform glands	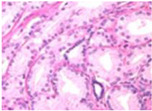	Intermediate-grade cancer
4 + 3	7	3	Intermediate	Poorly formed/fused/cribriform glands with lesser (5%) component of well-formed glands	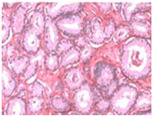	
4 + 4	8	4	High	Poorly formed/fused/cribriform glands	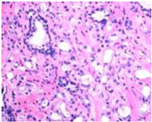	High-grade cancer
3 + 5				Well-formed glands with a few areas lacking glands (or with necrosis)	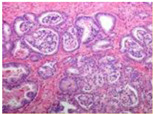	
5 + 3				Lack glands (or with necrosis) and a few well-formed glands	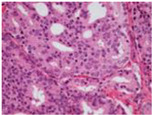	
4 + 5 5 + 4 5 + 5	9 or 10	5	Very high	Lack gland formation (or with necrosis) with or without poorly-formed/fused/cribriform glands	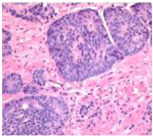	

**Table 2 biomedicines-11-03140-t002:** Involvement of different leukocytes and their associated cytokines in cancer and prostate cancer progression.

Leukocyte	Inflammatory Mediator Produced	Effect	Refs.
Neutrophils	MMP-9	MMP-9 produced by TANs and neutrophils degrade ECM leading to cancer progression in human xenografts and Mmp9-knockout mice	[104]
	GM-CSF and CXCL8	KRAS stimulated the expression of GM-CSF and CXCL8 in neutrophils which modulates the tumour microenvironment towards cancer progression in mouse models of ovarian cancer	[105]
	IL-8/CXCR2	Overexpression of CXCR2 in neutrophils promotes their attachment in lung cancer regions in a K-RAS mutant mouse model of lung cancer	[106]
	NDE, ROS, RNE	NDE, ROS, and RNE release from neutrophils lead to hMSH2-dependent G2/M checkpoint arrest and for the presence of replication errors in a co-culture model that mimics intestinal inflammation in ulcerative colitis	[107]
	ARG-1	Release of ARG-1 from neutrophils inhibit CD3-mediated T cell activation and proliferation leading to cancer progression in classic Hodgkin Lymphoma patients	[108]
	NET	Increased neutrophil and NET formation intended attenuate the rate of metastatic PCa in bones in vitro and an in vivo mouse model	[71]
	IL-1	IL-1RA from neutrophils leads to inhibition of senescence promoting cancer progression	[109]
	TNF-α	TNFR1 KO mice with depletion of TNF receptor 1 developed smaller tumours with attenuated proliferation and absence of metastasis	[110]
	Cathepsin G	Cathepsin G release from neutrophils increases cancer cell adhesion, and aggregation, and metastasis in breast cancer cells	[111]
Basophils	IL-4	Basophils from pancreatic ductal adenocarcinomas secrete IL-4 which induce GATA-3 expression in Th2 cells in patient samples and Mcpt8Cre mice	[112]
	CCL3	Basophils express CCL3 to negatively regulate the normal hematopoietic process in MCPT8-DTR mice and bone marrow samples from patients with CML	[113]
	CCL3/CCL4	Basophil plays a role in tumour rejection by increasing CD8^+^ T cell infiltration promoted by CCL3 and CCL4 in HCmel12-, B16-, and 616-OVA-induced transgenic FOXP3.LuciDTR-4 mice melanoma	[114]
	VEGFA	Immunologic activation by VEGF-2 of basophils induced the release of VEGF-A which induce basophil chemotaxis	[115]
	HGF	HGF is expressed in CML basophils in KU812-induced CML cell line	[116]
	ANGPT	Basophils express ANGPT1 and ANGPT2 mRNAs	[117]
Eosinophils	IL-2	IL-2 activate Tregs and Th17 cells involved in the promotion of cancer in a mouse model of PCa, and fibrosarcoma and head and neck human cancer tissues	[118]
	IL-4	IL-4 production promotes tumour growth and interaction with TAMs in a pancreatic-induced cancer mouse model	[78]
	IL-6	Increase of IL-6 correlated in patients with metastatic PCa compared with localized PCa	[119]
	IL-5 and CCL17	Eosinophils increase after Sipuleucel-T treatment of patients with metastatic castration-resistant PCa correlated with increase of IL-5 and CCL17, survival and maximal T-cell proliferation responses	[120]
	IFN-γ	IFN-γ induced CD4^+^ T cells to eliminate MHC II-negative cancer cells	[121]
	TNF-α	TNF-α correlated with increased extension of PCa in samples from PCa patients	[119]
	TGF-α	Overexpression of TGF-α decreased latency, increased growth, and tumour size of bladder cancer rat model	[122]
	VEGF	VEGF associated with poor prognosis of human small-cell lung carcinoma	[123]
	GM-CSF	Expression of GM-CSF correlated with NF-κB activation in bone-metastatic tumour tissues from individuals with metastatic breast cancer	[124]
Mast cells	Chymase	Chymase released from human mast cell release latent TGF-β-binding protein from the matrix	[125]
	Histamine	Histamine inhibition from mast cells inactivate EMT and cholangiocarcinoma growth via inhibition of c-Kit signalling in Mz-ChA-1-induced cholangiocarcinoma mouse model and human Mz-ChA-1 cells	[126]
	TNF-α	TNF-α released from mast cells amplifies and activates the functionality of CD8^+^ dendritic cells in Mcpt5-CreTNF^fl/fl^ mice	[127]
	IL-1β	Overexpression of IL-1β promoted tumour invasiveness and metastasis by inducing the expression of angiogenic genes and growth factors	[128]
	IL1, IL-4, IL-6	Decreased cell growth and participates in tumour rejection in breast cancer cells	[129]
	IL-8, IL-10	Mast cell-derived IL-8 and IL-10 act as tumour suppressors contributing to tumour cell growth	[130,131]
	PGD2	PGD2 secretion from mast cells attenuates angiogenesis in a Lewis lung carcinoma mouse model	[132]
Macrophages	IL-1β	IL-1β induced Snail stabilization in Snail/MCF7 cells and this effect was dependent on cell types and IL-1β concentration	[133]
	IL-8	IL-8 produced by macrophages induce EMT in hepatocellular carcinoma samples via JAK2/STAT3/Snail pathway	[134]
	TNF-α	TNF-α induces the stabilization of Snail in a non-phosphorylated, functional form and thus enhances cell migration and invasion dependent on NF-κB activation	[133]
	TGF-β	TGF-β induced EMT phenotypes in A549 cells, including changes in cell morphology and induction of mesenchymal marker expression in part by NF-κB signalling	[135]
	MMP-2, MMP-9	Macrophage-derived MMP-9 and MMP-2 related with fibrous capsule leading led to the migration and invasion of hepatocellular carcinoma cells in human samples	[136]
	CHI3L1	M2 macrophage-secreted CHI3L1 promoted metastasis of gastric and breast cancer cells in vitro and in vivo; CHI3L1 interaction with IL-13Rα2 upregulates MMPs	[137]
	IL-23/IL-17	Upregulation of IL-23 leads to tumour growth and progression and development of a tumoral IL-17 response which promote tumorigenesis in a mouse model of colorectal cancer	[138]
	IL-6	TAM-derived IL-6 highly expressed in Hepatocellular carcinoma patients, which is correlated with disease grades and tumour progression	[139]
	PDGF	PDGF release from macrophages mediates the recruitment of pericytes in human melanoma cell lines and OCM-1-induced melanoma mouse model	[140]
T cells	IL17-A	Inhibition of IL17-A release by Th17 cells prevent development of microinvasive PCa in mouse models	[141]
	IL-17	IL-17-producing T cells can promote PCa progression by enhancing inflammation and angiogenesis	[69]
	PD-1	A high percentage of CD8^+^ T cells express PD-1 in PCa samples, which impair an effective immune response by these cells	[44]
	IL-3	IL-3 expressed by T cells increase the recruitment of basophils and immune cells into the tumour microenvironment, which is linked with a poor survival	[112]
	IFN-γ	IFN-γ can enhance antigen presentation and contribute to immune surveillance in PCa	[142]
	TNF-α	TNF-α produced by activated T cells regulated apoptosis, angiogenesis, and inflammation in PCa	[143]
	TGF-β	TGF-β produced by T cells can suppress and promote tumour growth in PCa depending on the signal it receives	[144]
B cells	Lymphotoxin	Lymphotoxin lead to CXCL13/IKKa/STAT3/E2F1/BMI1 (RNF51) activation, ubiquitination of histones within PCa cell nuclei and proliferation of androgen-deprived PCa cells in castration-resistant PCa in mice	[145]
	TGF-β	Secretion of TGF-β by B-cells leads to anergy of CD8^+^ T cells	[146]
	IL-2	IL-2 and IL-4 produced by B cells regulate the Th2 memory responses to *Heligmosomoides polygyrus* (*Hp*) in chimeric mice lacking AID infected with *Hp*	[147]
	IL-6	Chimera’s mice with B cell lack IL-6 have impaired Th1 and Th17 responses to Salmonella	[148]
	GABA	B cell-derived GABA promotes monocyte differentiation into anti-inflammatory macrophages that secrete IL-10 and inhibit CD8^+^ T cell killer function in mice	[149]

ANGPT: Angiopoietin, ARG: Arginase, CCL: CC chemokine ligand, CML: Chronic myeloid leukaemia, CXCL: Chemokine (C-X-C motif) ligand, ECM: Extracellular matrix, EMT: Epithelial-mesenchymal transition, FOXP3: Forkhead box subfamily 3, GABA: γ-amino butyric acid, GM-CSF: Granulocyte-macrophage colony-stimulating factor, HGF: Hepatocyte growth factor, IFN-γ: Interferon-gamma, IL: Interleukin, IL-13Rα2: Interleukin-13 receptor α2 chain, IL-1RA: IL-1 receptor antagonist, MHC: Major histocompatibility complex, MMP: Matrix metalloproteinase, NDE: Neutrophil-derived elastase, NET: Neutrophil extracellular traps, PD: Programmed death, PDGF: Platelet-derived growth factor, PGD: Prostaglandin, RNE: Reactive nitrogen species, ROS: Reactive oxygen species, TAM: Tumour-associated macrophages, TAN: Tumour-associated neutrophils, TGF: Transforming growth factor, TNF: Tumour necrosis factor, VEGF: Vascular endothelial growth factor.

**Table 3 biomedicines-11-03140-t003:** Effect of polyphenols in prostate cancer.

Polyphenol	Model	Study Conditions	Effect	Ref.
Curcumin	PC-3 and DU-145 cells	0–50 μM curcumin, 0–48 h, 37 °C	5 μM curcumin reduced cell viability and proliferation in DU-145 cells; 25 μM curcumin reduced the survival and migration of DU-145 and PC-3 cells in 24 h	[250]
	PC-3 and DU-145 cells	10, 20, 30, 40 or 50 µmol/L curcumin at 37 °C for 12 h, 24 h or 48 h	30 µmol/L curcumin for 24 h decreased cell proliferation, migration, and invasion in PC-3 and DU-145 cells by regulating the miR-30a-5p/PCLAF axis	[251]
	LNCaP xenografts mice	OA of 30 mg/kg curcumin 3×/week in athymic nude mice injected s.c with LNCaP cells	Increased TUNEL staining, decreased the expression of PCNA andKi67, and inhibition of the activation of NF-kB in LNCaP xenografts	[252]
	Immunodeficient mice	S.c injection of PC-3 cells and daily i.p injected after 4–6 weeks with curcumin analogues (10 μg/g body weight) for 31 days	Curcumin analogues inhibited growth and progression of PC-3 tumours	[253]
	BALB/c-nu/nu	PC-3 cells injected s.c in mice and 0.25 μmol Ca 37, 0.5 μmol Ca 37, or 6 μmol curcumin administered i.p daily for 16 days	Ca 37 analogue suppressed PCa tumour and promoted curcumin-induced growth inhibition of PCa cells	[254]
Anacardic acid	C57BL/6 mice and nude mice	PC-3 cells s.c injected into mice and anacardic acid (2 mg/kg per day) s.c injected for 30 days	Inhibition of VEGF-induced cell proliferation, migration, and adhesion	[255]
	LNCaP cells	1–125 µmol/L anacardic acid at 30 °C for 24 h	125 µmol/L anacardic acid inhibited LNCaP cell proliferation, induced G1/S cell cycle arrest and apoptosis of LNCaP cells	[256]
Caffeic acid	PC-3, DU-145 and LNCaP cells	10–10^6^ nM CAPE for 72 h	CAPE attenuates proliferation and promotes and cytotoxic effect by reducing AKT, ERK and ER-a(Ser-167) phosphorylation in PC-3 cells	[257]
	PC-3 cells	20 µM CAPE for 24 h or 72 h	CAPE decreased protein expression of cyclin D1, cyclin E, SKP2, c-Myc, AKT, mTOR, and Bcl-2	[258]
Ellagic acid	LNCaP cells	25 and 50 μM ellagic acid for 48 h	Increased ROS, TGF-β, IL-6, and tumour suppressor protein p21 levels and activated caspase-3	[259]
	PC-3 and PLS-10 cells	0, 25, and 50 μM ellagic acid for 24 h, 37 °C	Decreased secretion of MMP-2 and proteolytic activity of collagenase/gelatinase secreted from PLS-10, inhibiting invasiveness of PCa cells	[260]
Gallic acid	DU-145 cells	24 h, 48 h, or 72 h	100 mg/mL gallic acid promoted maximal growth inhibition at 72 h and 25 and 50 mg/mL maximal apoptotic death at 24 and 48 h in human DU-145 cells	[261]
	DU-145 and 22Rv1 xenograft mice	Mice supplemented with 0.3% or 1% (*w*/*v*) gallic acid	Gallic acid feeding inhibited the growth of DU-145 and 22Rv1 PCa xenografts in nude mice	[262]
Resveratrol	Immunodeficient (SCID) mice	C-3M-MM2 cells s.c injected, and 20 mg/kg resveratrol administered oral gavage every 2 days	Inhibited PCa growth and metastatic lung lesions associated with reduced miR-21 and pAKT, and elevated PDCD4 levels	[263]
	BALB/cAnNCr-*nu/nu* mice	Supplemented with 50 and 100 mg/kg resveratrol. 2 weeks after LNCaP cells s.c injected	Delayed LNCaP tumour growth and inhibited expression of a marker for steroid hormone responses	[264]
	TRAMP-C1, TRAMP-C2, and TRAMP-C3 cells	50 or 100 μM resveratrol for 0 h, 2 h, 4 h, 8 h, 12 h, and 16 h	TRAMP cells exposed to resveratrol showed mitochondria-mediated decreased cell viability, and altered cell morphology leading to aberrant expression of Bax and Bcl-2 proteins	[265]
	PC-3 and 22RV1 cells	2.5–10 μM resveratrol	Resveratrol arrested cell cycle, promoted apoptosis, and sensitized PCa cells to ionization therapy, activated the ATM-AMPK-p53-p21cip1/p27kip1 and inhibit the AKT signaling pathways	[266]
Piceatannol	DU-145, MLL, PC-3 and TRAMP-C2 cells	Treatment with EGF for 0 h, 6 h, 12 h or 24 h with 0 or 10 μmol/L piceatannol	Piceatannol reduced basal and EGF-induced migration and invasion of DU-145 cells-induced IL-6 secretion by IL-6/STAT3 inhibition	[267]
	DU-145 cells	0 or 40 μmol/L piceatannol for 24 h	Piceatannol increased the percentage of cells in G1 phase, cyclin A, cyclin D1, and reduced CDK4 and CDK2 activity	[268]
	DU-145 cells	0–40 μM piceatannol for 24 h	Piceatannol reduced TNF-α-induced invasion and MMP-9 gene expression via suppression of NF-κB activity	[269]
Pterostilbene	PC-3 and LNCaP cells	0.1, 1, 10, 100, and 1000 mM pterostilbene for 24 h at 37 °C	A conjugate molecule caused 50% growth inhibition, reduced accumulation of cells in G2/M phase and induction of apoptosis by downregulation of PI3K/AKT and MAPK/ERK pathways	[270]
	PC-3 and LNCaP cells	0, 20, 40, 60, 80, and 100 mM pterostilbene for 48 h	80 μM pterostilbene decreased lipid synthesis by decreasing FASN expression and inhibiting ACC activity, blocked cell cycle at G1 phase by inducing p53 and further up-regulating p21 expression	[271]
Epigallocatechin-3-gallate (EGCG)	PC-3 cells	1 and 25 μM EGCG for 48 h	1 µM EGCG reduced PC-3 cell survival, promoted apoptosis by increasing the pro-apoptotic splice isoform of caspase-9 and enhanced apoptotic capacity of cisplatin	[272]
	DU-145, PC-3 and LNCaP cells	20, 40, 80, and 100 μg/mL EGCG for 24 h	EGCG inhibited cytokine and chemokine gene induction, activity of MMP-9 and -2, and NF-κB activity	[273]
Fisetin	LNCaP cells	Treatment with fisetin 10–60 μM, 48 h	Fisetin induced apoptosis, PARP cleavage, modulation of Bcl-2 family protein expression, inhibition of PI3K, phosphorylation of AKT at Ser473 and Thr308, mitochondrial release of cytochrome c into cytosol, and activation of caspases-3, -8 and -9	[274]
	LNCaP cells	Treatment with fisetin 10–60 μmol/L 48 h	Fisetin acted as an AR ligand leading to decrease in AR stability and decreased transactivation of target genes including PSA	[275]
	PC-3, DU-145 and LNCaP cells	Cells treated with fisetin 10–120 μM for 24 h, 48 h, 72 h, and 96 h	Fisetin activated the mTOR repressor TSC2 through inhibition of AKT and activation of AMPK leading to inhibition of Cap-dependent translation and induction of autophagic cell death in PC-3 cells	[276]
Quercetin	Cancer-induced (MNU and Testosterone treated) rats	Rats treated orally with 200 mg/kg quercetin 3×/week	Quercetin decreased expression of IGFIR, AKT, AR, cell proliferative and anti-apoptotic proteins	[277]
	LNCaP, DU-145, and PC-3 cells	5, 10, 20, 40, 80, and 160 μM quercetin for 24 h, 48 h, and 72 h	Apoptotic and necrotic cell death and AKT and NF-κB activation in PC-3 and LNCaP cells; reduction of AKT pathway and activation of Raf/MEK in DU-145 cells	[278]
	LNCaP cells	0, 1, 10, 25, 50, 100, and 150 μM quercetin for 24 h to 5 days depending on the type of analysis	Inhibited expression and function of the AR in LNCaP cells, decreased mRNA levels of *PSA, NKX3.1*, and *ODC* and repressed AR expression	[279]
	LNCaP cells	50–200 μM quercetin for 24 h and 48 h	Quercetin at 150 μM caused G0/G1 phase arrest via decreasing the levels of CDK2, cyclins E, and D proteins, stimulated the protein expression of ATF, GRP78, and GADD153, apoptotic cell death and DNA damage at 48 h, decreased Bcl-2, increased Bax, and activation of caspase-3, -8, and -9	[280]
	BALB/cA nude mice	PC-3 cells injected s.c, 20 mg/kg/d quercetin injected i.p for 16 days	Reduced tumour growth, inhibited tumorigenesis by targeting angiogenesis, reduced cell viability and induced apoptosis correlated with downregulation of AKT, mTOR and P70S6K	[281]
Apigenin	PC-3 xenograft mice	Orally administration of 20 and 50 μg/mouse/day apigenin for 8 weeks	Both doses of apigenin decreased tumour growth, HDAC activity, HDAC1 and HDAC3 protein expression, and bcl-2 expression shifting the Bax/Bcl-2 ratio in favour of apoptosis	[282]
	LNCaP and PC-3 cells	0, 10, 20, 40, and 80 μM apigenin for 72 h at 37 °C	Up-regulation of p21 expression, and p21 inhibits transcription of PLK-1	[283]
	PC-3 and DU-145 cells	5–40 μM apigenin for 24 h	Dose-dependent suppression of XIAP, c-IAP1, c-IAP2 and survivin protein levels, decrease in cell viability and apoptosis, decrease in Bcl-xL and Bcl-2, and inhibition of class I histone deacetylases and HDAC1 protein expression	[284]
	C57BL/TGN TRAMP mice	20 and 50 µg/day of apigenin for 20 weeks	Inhibition of VEGF, uPA, MMP-2, and MMP-9, tumour growth, and metastasis, reduction of IGF-I, and increase of IGFBP-3 through inhibition of p-AKT and p-ERK1/2	[285]
	TRAMP mice	20 and 50 μg/mouse/day, 6 days/week for 20 weeks	Apigenin-treated mice showed reduced proliferation, reduced phosphorylation of AKT (Ser473) and FoxO3a (Ser253), and upregulation of FoxO-responsive proteins BIM and p27/Kip1	[286]
Proanthocyanidins	DU-145 cells	0.1, 0.5 and 1.0 mg/mL PAC for 24 h	Down-regulation of MMP activity and up-regulation of TIMP activity	[287]
	DU-145 and LNCaP cells	20 μg/mL of blueberry fraction, 2.38 mM quercetin for 48 h (DU-145) or 72 h (LNCaP)	Inhibited growth of DU-145 and LNCaP cells	[288]
	DU-145, PC-3, and LNCaP cells	1–100 mg/mL PAC complex for 48 h	Inhibited proliferation of PC-3 and DU-145 with higher effect in LNCaP cells by decreasing AR expressing, G1 cell cycle arrest, decreased cyclin-dependent kinases, and cyclins, stimulated p21 and p27, and increased phosphorylation of p44 and p42	[289]

ACC: Acetyl-CoA carboxylase, AR: Androgen receptor, CAPE: Caffeic acid-phenyl ester, CDK: cyclin-dependent kinase, EGF: Epidermal growth factor, FASN: Fatty acid synthase, HPLC: High-performance liquid chromatography, I.p: Intraperitoneally, OA: Oral administration, PAC: Proanthocyanidin complex, PARP: Poly (ADP-ribose) polymerase, PI3K: Phosphatidyl inositol 3-kinase, PSA: Prostate-specific antigen, S.c: Subcutaneously, TRAMP: Transgenic adenocarcinoma of mouse prostate.

**Table 4 biomedicines-11-03140-t004:** Effect of polyphenol-based nanoparticles in prostate cancer.

Nanoparticle (Size)	Model	Treatment	Effect	Ref.
Curcumin-CA-NP (12.53–60.23 nm)	DU-145 cells	0.0832–260 μM 72 h	Uptake by PCa cells	[299]
Curcumin emulsome NP (184.21 nm)	LNCaP cells	10–40 μM for 24 h, 48 h or 72 h	Decreased proliferation, cell cycle arrest at G2/M phase	[300]
Curcumin NPs (34.0–359.4 nm)	PC-3 cells	50–600 μM overnight	Decreased cell viability and increased haemolytic effect	[301]
FA-RES + DTX-PBM NP (36.6 nm)	PC-3, C4-2B and LNCaP cells	3 μM RES + 0.01 μM DTX 24 h, 48 h, or 72 h	Reduced expression of NF-kB p65, COX-2, pro- and anti-apoptotic genes	[302]
RSV-SLN (126.85 nm)	PC-3 cells and Charles Foster rats	2 mL for 0–48 h and 2 mg/kg i.v. for 0–24 h	Internalization of NPs in PC-3 cells	[303]
RES-MSNs (60 nm)	PC-3 cells	10–20 μg for 72 h	Increased anti-proliferative activity and sensitization of Docatexal	[304]
RL-loaded PLGA (202.8 nm)	LNCaP cells	0–50 μM for 48 h	Decreased cell viability, G1-S phase arrest, increased apoptosis	[305]
EGCG-PA-PEG-NP (N.D.)	Tumour xenograft mice	1 mg in food consumption	Proapoptotic and angiogenesis inhibitory effects, enhanced bioavailability	[306]
^198^AuNP-EGCg (535 nm)	PC-3 xenograft SCID mice	136 μCi I.T. for 42 days	72% retention in tumours after 24 h and 80% reduction of tumour volumes after 28 days	[307]
EGCG-GA-MD-NPs (120 nm)	DU-145 cells	0.9–60 mg/mL for 64 h	Reduced cell viability and apoptosis induction	[308]
Chitosan-based EGCG NP (150–200 nm)	Athymic nude xenograft mice	3–6 mg/kg by O.A 5x week	Decreased tumour growth and PSA levels	[309]
EGCG-gold NPs (90.3 nm)	PC-3 cells	0–200 μg/mL for 1–24 h	Increased NF-κB activity and apoptosis	[310]

^198^AuNP-EGCg: Epigallocatechin-gallate functionalized radioactive gold nanoparticles, FA-RES + DTX-PBM NP: Folic acid conjugated resveratrol and docetaxel planetary ball milled nano-particle, Curcumin-CA-NP: Curcumin loaded calcium alginate nanoparticles, EGCG-GA-MD-NPs: Epigallocatechin-gallate-based gum arabic maltodextrin nanoparticles, EGCG-PA-PEG-NP: Epigallocatechin-3-gallate encapsulated polylactic acid–polyethylene glycol nanoparticles, i.t.: Intratumorally, i.v.: Intravenously, N.D: Not defined, O.A: Oral intubation, PSA: Prostate specific antigen, RES-MSNs: Resveratrol-based mesoporous silica nanoparticles, RL-loaded PLGA: Resveratrol-loaded polylactic-co-glycolic acid, RSV-SLN: Resveratrol-solid lipid nanoparticles.

## Data Availability

Not applicable.

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
