# Peer review of "Inflammation in Prostate Cancer: Exploring the Promising Role of Phenolic Compounds as an Innovative Therapeutic Approach"

_biomedicines, 2023, doi:10.3390/biomedicines11123140_

Round 1
Reviewer 1 Report
Comments and Suggestions for Authors
Reviewer comments and suggestions
The authors in this study discuss the diagnosis methods for Prostate cancer and the pathophysiological mechanisms underlying the disease, illuminating the dynamic interplay between inflammation and leukocytes in promoting tumor development and spread. They have discussed the polyphenols, with their noteworthy antioxidant and anti-inflammatory properties, along with their synergistic potential when combined with conventional treatments, offer promising prospects for innovative therapeutic strategies. Moreover, they have mentioned the evidence from the use of polyphenol and polyphenol-based nanoparticles in PCa revealed their positive effects in controlling tumour growth, proliferation, and metastasis.
Overall, the manuscript is well written. I have listed the concerns and comments that needed to be explained or modified.
- Line 41 Do you have any appropriate reference, Kindly mention
- Line 53-55 What were the differences, as both are mentioning about immune cells?
- Line 60-61 Please cite a previous published manuscript
- Comments for figure 1 If you have adapted not redrawn then you need to take copyright. Have to take permission
- Similar comments as of above for the table 1 (related to the figures used)
- Line 240 a typo error was found in “imaging”
- Please cite more references for line 268 as the authors suggested studies have demonstrated
- Line 315 This section was really big, I think not necessary to discuss all included (already results were included in table); instead the authors could explore more on the therapeutic part
- Comments for Table 3 The authors can categorize one more column for models they have used and can explore more on nanoparticles used by the different authors
- Section 7 could be elaborated more for the common reader of your manuscript
- Section 8.1 The section briefly discussed the need to touch other nanoparticles that were used for treating prostate cancer
- Please check the reference 14, 25, 35, 39 and many more. Please proofread again.
Author Response
Dear Editor of Biomedicines,
In reply to the review performed on the paper entitled “Inflammation in Prostate Cancer: Exploring the promising role of Phenolic Compounds as an innovative therapeutic approach”, we would like to acknowledge the valuable comments performed by the editor that kindly accepted to revise our manuscript. We would like to confirm that we have addressed most issues and answered the questions made by reviewer 1 (all the modifications are highlighted in yellow). We hope the answers below and modifications that have been done in the manuscript are clear and concise enough as required by the reviewer to enable the publication of the manuscript in Biomedicines.
Answer to referee’s comments and queries
Detailed responses to Reviewer 1
Reviewer´s comment: Line 41 Do you have any appropriate reference, Kindly mention
Our reply: Thank you for the comment. The references related to the information were added in line 41.
Reviewer´s comment: Line 53-55 What were the differences, as both are mentioning about immune cells?
Our reply: Thank you very much for the comment. What we aim to highlight is the notable gap in existing review articles, which predominantly address the involvement of the most common immune cells—macrophages, T cells, B cells, and neutrophils—and not other potential immune cells such as basophils and eosinophils. Other review articles often delve into the general role of inflammatory mediators without specifying the origin or contribution of a particular mediator from a specific inflammatory cell. This review article distinguishes itself by providing an in-depth exploration of each inflammatory cell's role and, additionally, by describing the inflammatory mediators produced by each, thereby highlighting the individual contribution of each cell type to cancer progression.
Reviewer´s comment: Line 60-61 Please cite a previous published manuscript
Our reply: Thank you for the comment. The references in line 61 were added as suggested.
Reviewer´s comment: Comments for figure 1 If you have adapted not redrawn then you need to take copyright. Have to take permission
Our reply: Thank you for the comment. In fact, the figure was redrawn for this review. The images were constructed by us, and the information contained in the images are also different from the reference described.
Reviewer´s comment: Similar comments as of above for the table 1 (related to the figures used)
Our reply: Thank you for the question. This table was also created by us only with information of the described references as explained in the previous question.
Reviewer´s comment: Line 240 a typo error was found in “imaging”
Our reply: Thank you for the comment that it was already corrected.
Reviewer´s comment: Please cite more references for line 268 as the authors suggested studies have demonstrated
Our reply: Thank you for the consideration. The references were added as suggested.
Reviewer´s comment: Line 315 This section was really big, I think not necessary to discuss all included (already results were included in table); instead the authors could explore more on the therapeutic part
Our reply: Thank you for the comment. The section was revised as suggested.
Reviewer´s comment: Comments for Table 3 The authors can categorize one more column for models they have used and can explore more on nanoparticles used by the different authors
Our reply: Thank you for your suggestion. We incorporated an additional column for models in Table 3 and introduced a new table (Table 4) specifically dedicated to nanoparticles used by different authors. This separation ensures a clear and organized presentation, preventing the mixing of different data and information.
Reviewer´s comment: Section 7 could be elaborated more for the common reader of your manuscript
Our reply: Thank you for the comment. Reviewer 2 has suggested to summarize the content of section 7 and include it on section 7 to focus the main topic of the manuscript: inflammation and polyphenols. This way, we have improved this section has suggested but, in a way, to not turn the manuscript quite long and unfocused.
Reviewer´s comment: Section 8.1 The section briefly discussed the need to touch other nanoparticles that were used for treating prostate cancer
Our reply: In an industrial symbiosis framework, the imperative to replace conventional synthesis methods emerges as a critical requirement in reducing environmental impact. Consequently, this section distinctly highlights the pressing demand to substitute conventional nanoparticles with environmentally friendly alternatives, leveraging the potential of polyphenols. Polyphenols can act as reducing, stabilizing, and capping agents for nanoparticle allowing the replace of conventional nanoparticle synthesis, which uses toxic chemicals and metals, being highly pollutant for the environment and toxic.
Reviewer´s comment: Please check the reference 14, 25, 35, 39 and many more. Please proofread again.
Our reply: Thank you for the comment. All references were checked and corrected.
Sincerely,
Ana Isabel Ramos Novo Amorim de Barros

Reviewer 2 Report
Comments and Suggestions for Authors
Authors have reviewed the role of inflammation on PCa development and progression, as well as the potential role of polyphenols on disease management. The manuscript is well written and exhaustively documented. I have minor suggestions prior to publication:
1. Sections 4 and 7 should be extremely summarized. Despite well written and referenced, both sections fall out of the main topic of the manuscript: inflammation and polyphenols. I suggest that the main points of both of them should be included on section 2, given that both diagnosis and treatment are strongly correlated with prognosis. Otherwise, the manuscript results quite long and its main topic, unfocused.
2. On lines 291-292, authors mention that 'The link between a high-fat diet and obesity as risk factors for PCa development remains controversial.', whereas on lines 40-42 they state that 'With an aging population and the increasing prevalence of food processing and poor dietary habits, there is an anticipated rise in the absolute number of PCa cases.' Given that more than one factor (in this case, diet) influences the risk of PCa onset, I think authors should be careful on the Introduction when stating that poor diet will be the only (or the main) cause of an increase in PCa cases.
3. I noticed a minor misspelling on line 283: 'pf' instead of 'of'.
Author Response
Dear Editor of Biomedicines,
In reply to the review performed on the paper entitled “Inflammation in Prostate Cancer: Exploring the promising role of Phenolic Compounds as an innovative therapeutic approach”, we would like to acknowledge the valuable comments performed by the editor that kindly accepted to revise our manuscript. We would like to confirm that we have addressed most issues and answered the questions made by reviewer 2 (all the modifications are highlighted in yellow). We hope the answers below and modifications that have been done in the manuscript are clear and concise enough as required by the reviewer to enable the publication of the manuscript in Biomedicines.
Answer to referee’s comments and queries
Detailed responses to Reviewer 2
Reviewer´s comment: Sections 4 and 7 should be extremely summarized. Despite well written and referenced, both sections fall out of the main topic of the manuscript: inflammation and polyphenols. I suggest that the main points of both of them should be included on section 2, given that both diagnosis and treatment are strongly correlated with prognosis. Otherwise, the manuscript results quite long and its main topic, unfocused.
Our reply: Following a thorough discussion on these topics, we have concluded that inserting the sections on diagnosis and current therapies before explaining the pathophysiology of the disease may not be the most optimal way to organize the information. Taking your suggestion into consideration, we have reorganized and summarized the sections in the following sequence: First, "Epidemiology of Prostate Cancer," followed by "Diagnosis of Prostate Cancer," then "Pathophysiology of Prostate Cancer," and finally, the section on Current Therapeutic Strategies used in Prostate Cancer.
Reviewer´s comment: On lines 291-292, authors mention that 'The link between a high-fat diet and obesity as risk factors for PCa development remains controversial.', whereas on lines 40-42 they state that 'With an aging population and the increasing prevalence of food processing and poor dietary habits, there is an anticipated rise in the absolute number of PCa cases.' Given that more than one factor (in this case, diet) influences the risk of PCa onset, I think authors should be careful on the Introduction when stating that poor diet will be the only (or the main) cause of an increase in PCa cases.
Our reply: Thank you for your comment. The introduction (lines 40-45) was improved according to these suggestions.
Reviewer´s comment: I noticed a minor misspelling on line 283: 'pf' instead of 'of'.
Our reply: Thank you for the correction that was already altered. The references in line 61 were added as suggested.
Sincerely,
Ana Isabel Ramos Novo Amorim de Barros

Round 2
Reviewer 1 Report
Comments and Suggestions for Authors
No more comments. Thank you